# 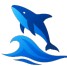 HiFlow: Training-free High-Resolution Image Generation with Flow-Aligned Guidance

**Jiazi Bu**[1,5*]  **Pengyang Ling**[2,5*]  **Yujie Zhou**[1,5*]  **Pan Zhang**[5†]  **Tong Wu**[4]
**Xiaoyi Dong**[3,5]  **Yuhang Zang**[5]  **Yuhang Cao**[5]  **Dahua Lin**[3,5,7]  **Jiaqi Wang**[5,6†]
[1]Shanghai Jiao Tong University  [2]University of Science and Technology of China
[3]The Chinese University of Hong Kong  [4]Stanford University  [5]Shanghai AI Laboratory
[6]Shanghai Innovation Institute  [7]CPII under InnoHK
https://bujiazi.github.io/hiflow.github.io/

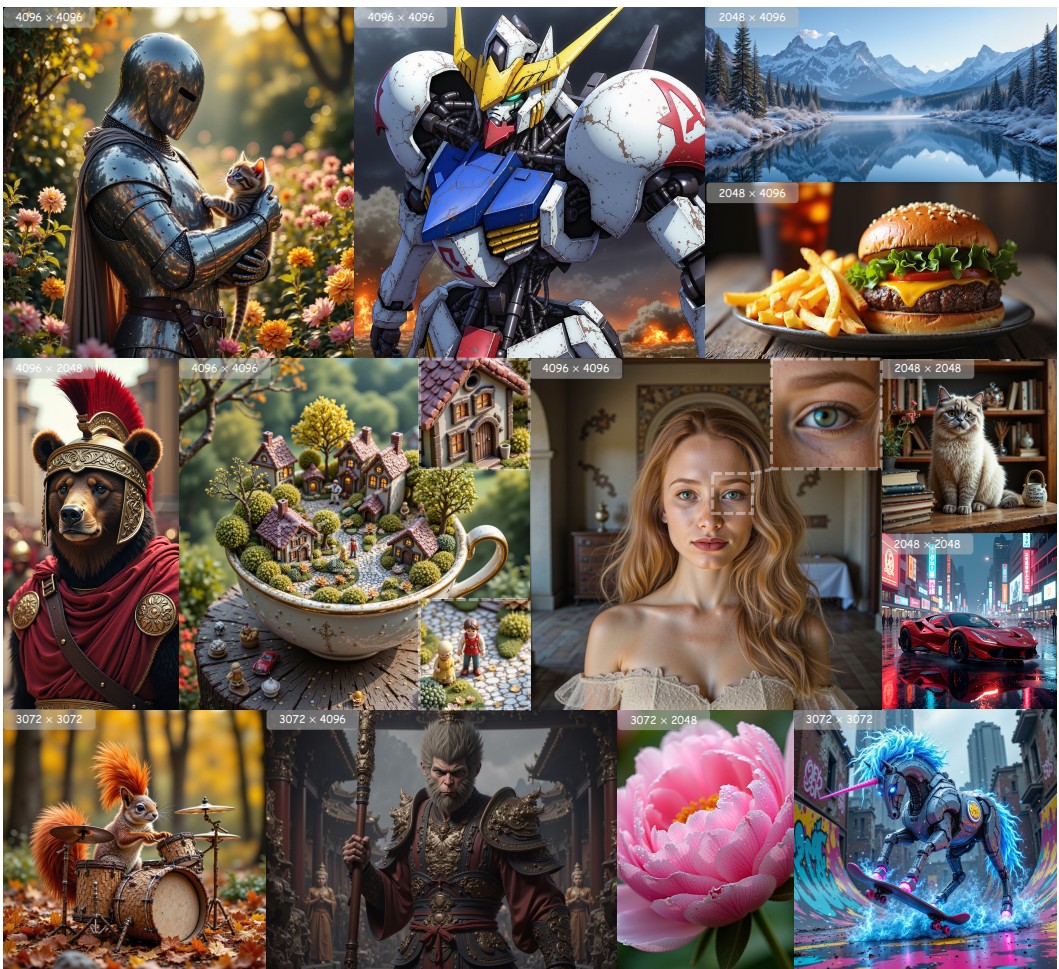

Figure 1: **Gallery of HiFlow.** The proposed HiFlow enables pre-trained text-to-image flow models (Flux.1.0-dev integrated with various LoRA models) to synthesize high-resolution images with high fidelity and rich details in a training-free manner. **All prompts are listed in the appendix.**

---

*Equal contribution. †Corresponding author.

39th Conference on Neural Information Processing Systems (NeurIPS 2025).

## Abstract

Text-to-image (T2I) diffusion/flow models have drawn considerable attention recently due to their remarkable ability to deliver flexible visual creations. Still, high-resolution image synthesis presents formidable challenges due to the scarcity and complexity of high-resolution content. Recent approaches have investigated training-free strategies to enable high-resolution image synthesis with pre-trained models. However, these techniques often struggle with generating high-quality visuals and tend to exhibit artifacts or low-fidelity details, as they typically rely solely on the endpoint of the low-resolution sampling trajectory while neglecting intermediate states that are critical for preserving structure and synthesizing finer detail. To this end, we present **HiFlow**, a training-free and model-agnostic framework to unlock the resolution potential of pre-trained flow models. Specifically, HiFlow establishes a virtual reference flow within the high-resolution space that effectively captures the characteristics of low-resolution flow information, offering guidance for high-resolution generation through three key aspects: initialization alignment for low-frequency consistency, direction alignment for structure preservation, and acceleration alignment for detail fidelity. By leveraging such flow-aligned guidance, HiFlow substantially elevates the quality of high-resolution image synthesis of T2I models and demonstrates versatility across their personalized variants. Extensive experiments validate HiFlow's capability in achieving superior high-resolution image quality over state-of-the-art methods. Our code is available at HiFlow Repo.

## 1   Introduction

The text-to-image (T2I) diffusion/flow models [45, 41, 40, 4, 10, 38, 40, 46, 30, 50, 28, 53, 54, 3], which allow for flexible content creation from textual prompts, have recently achieved a landmark advancement. Despite considerable improvements, existing T2I models are typically confined to a restricted resolution (e.g., $1024 \times 1024$) and experience notable quality decline and even structural breakdown when attempting to generate higher-resolution images, as illustrated in Fig. 2. Such shortcoming limits the utility and diminishes their appeal to contemporary artistic and commercial applications, in which detail and precision are paramount.

As initial efforts, several methods [16, 21, 35, 44, 51, 63, 57, 56] suggest fine-tuning T2I models on higher-resolution samples to enhance the adaptability to large-scale images. Nevertheless, this straightforward approach entails significant costs, primarily the burden of high-resolution image collection and the necessity for model-specific fine-tuning. Therefore, recent studies have investigated training-free strategies [17, 12, 23, 1, 25, 60, 29, 26, 42, 13, 49] to harness the inherent potential of pre-trained T2I models in high-resolution image synthesis. However, the majority of these methods [17, 23, 60, 42, 25] involve manipulating the internal features within models, exhibiting restricted transferability across architectures, such as applying methods tailored for U-Net architecture in DiT-based models. Another line of research [12, 26, 49, 13] suggests fusing the upsampled low-resolution images into the denoising target during high-resolution synthesis for structure guidance. Despite simplicity, these methods merely align high-resolution images predicted at different time steps with the upsampled low-resolution image from the final step as single guidance anchor, risking the introduction of artifacts due to distribution discrepancy, as shown in Fig. 3 (b). Moreover, they primarily emphasize structural guidance, leaving the potential of detail-oriented cues underexplored, resulting in suboptimal detail fidelity in the outputs, as illustrated in Fig. 3 (d).

In this work, we introduce a novel training-free and model-agnostic generation framework, termed **HiFlow**, which is designed to advance high-resolution T2I synthesis of Rectified Flow models and can be seamlessly extended to diffusion models by modifying the denoising scheduler. Specifically, HiFlow involves a cascade generation paradigm: First, a virtual reference flow is constructed in the high-resolution space based on the step-wise estimated clean samples of the low-resolution sampling flow. Then, during high-resolution synthesizing, the reference flow offers guidance in sampling initialization, denoising direction, and moving acceleration, aiding in achieving consistent low-frequency patterns, preserving structural features, and maintaining high-fidelity details, respectively. Such flow-aligned guidance from the sampling trajectory facilitates better merging of the structure synthesized at the low-resolution scale and the details synthesized at the high-resolution

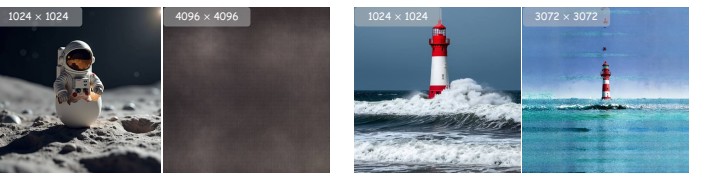

FLUX (DiT/Flow)  SD3 Medium (DiT/Flow)  SDXL (U-Net/Diffusion)

Figure 2: **T2I models suffer significant quality degradation in high-resolution image generation.**

scale, facilitating superior visual quality. Furthermore, HiFlow exhibits broad generalizability across U-Net and DiT architectures, owing to its independence from internal model characteristics. Extensive experiments demonstrate that the proposed method surpasses state-of-the-art baselines on the latest Rectified Flow T2I model. The main contributions of this paper are summarized as follows: (i) We propose **HiFlow**, a novel training-free and model-agnostic framework to unlock the resolution potential of pre-trained Rectified Flow models, which constructs virtual reference flow derived from low-resolution sampling trajectories to enable high-resolution synthesis; (ii) The constructed virtual reference flow provides flow-aligned guidance in terms of initialization, direction, and acceleration, thus promoting low-frequency consistency, structural preservation, and detail fidelity, respectively; and (iii) Comprehensive experiments validate HiFlow's superiority over state-of-the-art competitors and highlight its flexibility and versatility in various applications including LoRA [22], ControlNet [59], and Quantization [31].

## 2 Related work

### 2.1 Text-to-image generation.

Text-to-image (T2I) synthesis [45, 41, 40, 4, 10, 38, 46, 30, 50, 28] has witnessed significant advancements with the advent of diffusion models [2, 33, 64], which have demonstrated remarkable capabilities in producing high-quality images based on given textual prompts. Denoising Diffusion Probabilistic Models (DDPM) [19] and Guided Diffusion [9] showcased the potential of diffusion processes to generate high-fidelity images. Subsequently, the introduction of latent space diffusion [45] marked a revolutionary advancement in the field, which significantly reduced computational demands and enabled more efficient training, giving rise to pioneering models such as Stable Diffusion and Stable Diffusion XL [41]. Recently, the integration of transformer-based architectures [5, 4, 14, 15, 65, 28] into diffusion models has also led to improvements in both image quality and computational efficiency. In this work, we primarily construct our method upon Flux.1.0-dev [28], an advanced Rectified Flow T2I backbone renowned for its superior generation quality.

### 2.2 High-resolution image generation.

High-resolution image generation with diffusion models has gained increasing popularity in recent years. Several studies [16, 21, 35, 44, 51, 63, 57, 56] have proposed training or fine-tuning existing T2I diffusion models to enhance their capability for high-resolution image generation. However, it remains a challenging task due to the scarcity of high-resolution training data and the substantial computational resources required for modeling such data. Another line of researchs [58, 11, 32, 52, 6] employs super-resolution techniques to upscale the resolution of generated low-resolution images. Nevertheless, the quality of generated images via this approach heavily depends on the initial quality of the low-resolution images and the performance of the super-resolution model. Recent efforts have focused on training-free strategies [17, 12, 23, 1, 25, 60, 29, 26, 42, 13, 49, 55] that modify the inference strategies of diffusion models for low-resolution generation. For instance, HiDiffusion [60] suggests reshaping features in the outermost blocks of the U-Net architecture to match the training size in deeper blocks. DiffuseHigh [26] upscales the low-resolution generation result and re-denoises it under the structural guidance from the Discrete Wavelet Transform. I-Max [13] projects the high-resolution flow into the low-resolution space, enabling training-free high-resolution image generation with flow models. Many existing training-free methods [17, 23, 60, 42, 25] are inherently entangled with model-specific internal features, limiting their generalizability across different architectures. Others [12, 26, 49, 13] lack effective guidance throughout the high-resolution generation process,

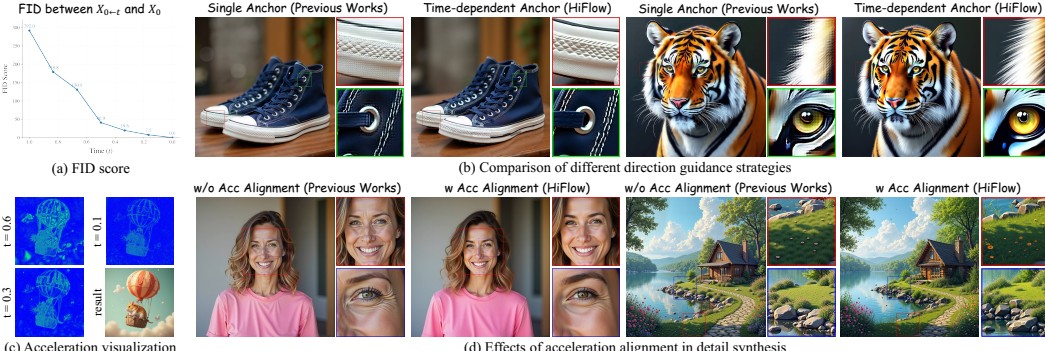

(a) FID score

(b) Comparison of different direction guidance strategies

(c) Acceleration visualization

(d) Effects of acceleration alignment in detail synthesis

Figure 3: **Observations.** (a) Distribution discrepancy between predicted clean sample $X_{0 \leftarrow t}$ and clean sample $X_0$. (b) Comparison with constant and time-dependent direction guidance. The former exhibits artifacts, the latter demonstrates better structure preservation. (c) Visualization of acceleration. (d) Effect of acceleration alignment, validating its role in facilitating high-fidelity details generation.

including exhibiting distributional discrepancy in the selection of structure guidance anchors and the neglect of detail synthesis guidance, resulting in artifact emergence and decreased detail fidelity.

## 3 Method

### 3.1 Preliminaries

Flow Matching [34] and Rectified Flow [36] aim at streamlining the formulation of Ordinary Differential Equation (ODE) models by establishing a linear transition between two distinct distributions. Consider clean image samples $X_0 \sim \pi_0$ and Gaussian noise $X_1 \sim \pi_1$. Rectified Flow delineates a linear trajectory from $X_1$ to $X_0$, with the intermediate state $X_t$ defined as:

$$X_t = tX_1 + (1-t)X_0, \tag{1}$$

in which $t \in [0,1]$ denotes a continuous time interval. By taking the derivative of $t$ on both sides of Eq. 1, its linear progression gives the following equation:

$$dX_t = (X_1 - X_0)dt. \tag{2}$$

During the denoising phase, given $X_t$ and time $t$, a neural network $v_\theta$ is introduced to estimate the vector of flow, i.e., $X_1 - X_0$), which can be expressed as:

$$v_\theta(X_t, t, c) = X_{1 \leftarrow t} - X_{0 \leftarrow t}, \tag{3}$$

in which $c$ represents optional control signs such as textual or image prompts, and $X_{1 \leftarrow t}$ and $X_{0 \leftarrow t}$ denote the predicted noisy and clear component within $X_t$, respectively. Further derivations regarding $X_{1 \leftarrow t}$ and $X_{0 \leftarrow t}$ are provided in the appendix. Since $X_t$ and $t$ are known, from Eq. 1, $v_\theta(X_t, t, c)$ is essentially determined by the predicted clean component $X_{0 \leftarrow t}$, i.e.,

$$v_\theta(X_t, t, c) = \frac{X_t - X_{0 \leftarrow t}}{t}. \tag{4}$$

The progressive denoising of flow models can be formulated as follows:

$$X_{t_{i-1}} = X_{t_i} + v_\theta(X_{t_i}, t_i, c)(t_{i-1} - t_i), \tag{5}$$

in which the movement of $t_i$ from 1 to 0 indicates the trajectory of $X_{t_i}$ from gaussian noise $X_1$ to clean sample $X_0$. During denoising, the predicted $v_\theta(X_{t_i}, t_i, c)$ determines the denoising direction at each time $t_i$. In the following sections, $v_t$ is used to denote $v_\theta(X_t, t, c)$ for simplicity.

### 3.2 Virtual reference flow

Rectified flow models have shown great promise in advancing high-quality image generations. Yet, these models experience a critical quality drop when attempting high-resolution synthesizing, as illustrated in Fig. 2. To this end, previous works [12, 26, 49, 13] typically leverage the synthesized

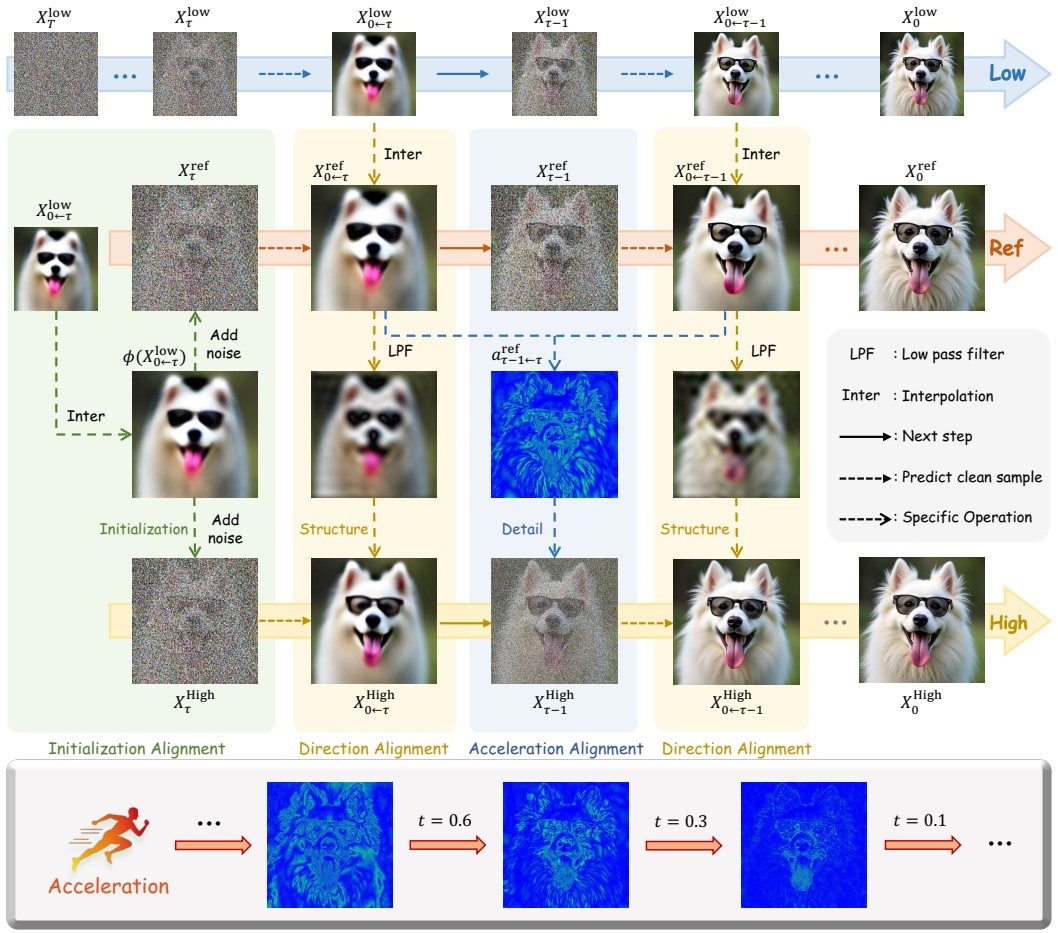

Figure 4: **Overview of HiFlow.** HiFlow constructs reference flow from low-resolution sampling trajectory to offer guidance for high-resolution generation in initialization, direction, and acceleration.

low-resolution sample $X_0^{\text{low}}$ for guidance by fusing its upsampled variant into predicted clean sample $X_{0 \leftarrow t}^{\text{high}}$ to modify the denoising direction. However, for given flow models, it is observed that there are ignored distribution discrepancies between $X_{0 \leftarrow t}$ at different $t$ and $X_0$, and these discrepancies increase significantly as $t$ approaches 1, as depicted in Fig. 3 (a). Therefore, the fusion between upsampled $X_0^{\text{low}}$ (single guidance anchor) and $X_{0 \leftarrow t}^{\text{high}}$ may cause artifacts and thus lead to sub-optimal results, as shown in Fig. 3 (b). To this end, in this work, *a virtual reference flow is constructed in the high-resolution space that can fully characterize the information of the low-resolution sampling trajectory.* Specifically, for reference flow, its predicted clean image $X_{0 \leftarrow t}^{\text{ref}}$ at time $t$ is defined as:

$$\left\{ X_{0 \leftarrow t}^{\text{ref}} \right\} = \left\{ \phi(X_{0 \leftarrow t}^{\text{low}}) \right\}, \quad t \in [0, 1] \tag{6}$$

in which $\phi(X_{0 \leftarrow t}^{\text{low}})$ is the upsampled $X_{0 \leftarrow t}^{\text{low}}$ by interpolation function $\phi(\cdot)$. Meanwhile, the noisy image $X_t^{\text{ref}}$ of the reference flow at time $t$ is defined as a virtual noisy image in high-resolution space. For each time $t$, the vector of reference flow is $(X_t^{\text{ref}} - X_{0 \leftarrow t}^{\text{ref}})/t = (X_t^{\text{ref}} - \phi(X_{0 \leftarrow t}^{\text{low}}))/t$. Indeed, the reference flow employs virtual $X_t^{\text{ref}}$ to construct an imaginary sampling trajectory in high-resolution space that can produce the upsampled low-resolution image $\phi(X_0^{\text{low}})$. Such a virtual reference flow acts as a bridge connecting low-resolution and high-resolution sampling trajectories, facilitating guided high-resolution synthesis with enhanced structure and fidelity.

### 3.3 Flow-aligned guidance

Given that T2I diffusion/flow models prioritize low-frequency structure components before synthesizing high-frequency details [50, 27], we follow the cascade generation pipeline [12, 26, 42], i.e., first synthesize a low-resolution image and mapping its sampling trajectory to high-resolution space to

obtain reference flow. This reference flow is then used for flow-guided high-resolution generation, which is analyzed from the following three aspects. The pipeline of HiFlow is illustrated in Fig. 4.

**Initialization alignment**. For a given virtual reference flow, the sampling of high-resolution generation starts from the noisy variant of $X_{0 \leftarrow \tau}^{\text{ref}}$, which can be expressed as:

$$X_\tau^{\text{high}} = \tau X_1^{\text{high}} + (1 - \tau) X_{0 \leftarrow \tau}^{\text{ref}}, \tag{7}$$

in which $X_\tau^{\text{high}}$ is the sampling initialization of high-resolution generation, $X_1^{\text{high}}$ is gaussian noise, and $\tau \in (0, 1)$ is noise addition ratio. Such initialization alignment allows skipping the early stage in high-resolution generation, thereby maintaining the consistency of high-resolution results and low-resolution images in low-frequency components, while also facilitating higher inference speeds.

**Direction alignment**. While initialization alignment ensures low-frequency consistency at the beginning of high-resolution generation, such structural information may be destroyed at subsequent denoising steps due to the limited stability of models in synthesizing high-resolution content (as shown in Fig. 2), risking broken structures. Therefore, direction alignment is designed for structural preservation, which is achieved by modifying the denoising direction based on reference flow, i.e.,

$$\hat{X}_{0 \leftarrow t}^{\text{high}} = X_{0 \leftarrow t}^{\text{high}} + \alpha_t [\widetilde{\mathcal{F}}(\mathcal{F}(X_{0 \leftarrow t}^{\text{ref}}) \odot \mathcal{L}(D)) - \widetilde{\mathcal{F}}(\mathcal{F}(X_{0 \leftarrow t}^{\text{high}}) \odot \mathcal{L}(D))], \tag{8}$$

in which $\mathcal{F}$ and $\widetilde{\mathcal{F}}$ denote 2D Fast Fourier Transform and its inverse operation, respectively, $\mathcal{L}(\cdot)$ denotes a butterworth low-pass filter and $D$ is the normalized cutoff frequency, and $\alpha_t$ is the direction guidance weight. Essentially, Eq. 8 replaces the low-frequency component of $X_{0 \leftarrow t}^{\text{high}}$ with that in $X_{0 \leftarrow t}^{\text{ref}}$, thus rejecting the updating on low-frequency structures when synthesizing high-frequency details. Unlike previous work [26, 12, 49] that aligns with a single guidance anchor $\phi(X_0^{\text{low}})$, Eq. 8 manipulates $X_{0 \leftarrow t}^{\text{high}}$ by using time-dependent $X_{0 \leftarrow t}^{\text{ref}}$, such fusion between samples from the same $t$ helps avoid artifacts caused by distribution discrepancy (see Fig. 3 (a)), facilitating better quality.

**Acceleration alignment**. Although the above strategies allow for rich detail generation while maintaining structure, the fidelity of synthesized details risks dropping in some cases, in which unrealistic contents appear, such as repetitive patterns and abnormal textures, as shown in Fig. 3(d). This issue primarily arises from the constrained capability of generation models in synthesizing high-resolution content. To this end, acceleration alignment is proposed to enhance detail fidelity by aligning the synthesized content of the high-resolution flow with the reference flow at each time $t$. Specifically, the acceleration, which is defined as the second-order derivative of movement $X_t$, and also denotes the first-order derivation of vector $v_t$, can be expressed as:

$$a_{t_{i-1} \leftarrow t_i} = \frac{d^2 X_t}{dt^2} = \frac{dv_t}{dt} = \frac{v_{t_{i-1}} - v_{t_i}}{t_{i-1} - t_i}. \tag{9}$$

Furthermore, based on Eq. 4 and Eq. 5, the acceleration in Eq. 9 can be simplified as follows:

$$
\begin{aligned}
a_{t_{i-1} \leftarrow t_i} &= \frac{\frac{1}{t_{i-1}}(X_{t_{i-1}} - X_{0 \leftarrow t_{i-1}}) - \frac{1}{t_i}(X_{t_i} - X_{0 \leftarrow t_i})}{t_{i-1} - t_i} \\
&= \frac{t_i X_{t_i} + (t_{i-1} - t_i)(X_{t_i} - X_{0 \leftarrow t_i}) - t_i X_{0 \leftarrow t_{i-1}} - t_{i-1}(X_{t_i} - X_{0 \leftarrow t_i})}{t_{i-1} t_i (t_{i-1} - t_i)} \\
&= -\frac{1}{t_{i-1}} \frac{X_{0 \leftarrow t_{i-1}} - X_{0 \leftarrow t_i}}{t_{i-1} - t_i}.
\end{aligned}
\tag{10}
$$

It can be concluded from Eq. 10 that the acceleration depicts the variation in the predicted clean sample $X_{0 \leftarrow t}$ between adjacent time $t$, with time-dependent term $1/t$. As shown in Fig. 3 (c), the acceleration primarily captures texture and contour information while also indicating the sequence of content synthesis at each $t$, i.e., it showcases what content the model is responsible for adding at different $t$. Therefore, we propose aligning the acceleration of high-resolution generation with that of the reference flow to synchronize the model's preference for content synthesis order, enabling guided detail synthesis both in content and timing. Mathematically, acceleration alignment is modeled as:

$$\hat{a}_{t_{i-1} \leftarrow t_i}^{\text{high}} = a_{t_{i-1} \leftarrow t_i}^{\text{high}} + \beta_t (a_{t_{i-1} \leftarrow t_i}^{\text{ref}} - a_{t_{i-1} \leftarrow t_i}^{\text{high}}), \tag{11}$$

in which $\beta_t$ is the acceleration guidance weight. Substituting Eq. 9 into Eq. 11, we have

$$\frac{\hat{v}_{t_{i-1}}^{\text{high}} - v_{t_i}^{\text{high}}}{t_{i-1} - t_i} = \frac{v_{t_{i-1}}^{\text{high}} - v_{t_i}^{\text{high}}}{t_{i-1} - t_i} + \beta_t \left( \frac{v_{t_{i-1}}^{\text{ref}} - v_{t_i}^{\text{ref}}}{t_{i-1} - t_i} - \frac{v_{t_{i-1}}^{\text{high}} - v_{t_i}^{\text{high}}}{t_{i-1} - t_i} \right). \tag{12}$$

Furthermore, Eq. 12 can be simplified into the following form:

$$\hat{v}_{t_{i-1}}^{\text{high}} = v_{t_{i-1}}^{\text{high}} + \beta_t(v_{t_{i-1}}^{\text{ref}} - v_{t_i}^{\text{ref}} - v_{t_{i-1}}^{\text{high}} + v_{t_i}^{\text{high}}). \tag{13}$$

Subsequently, the obtained $\hat{v}_t^{\text{high}}$ based on Eq. 13 is plugged into the $v_\theta$ in Eq. 5 to facilitate acceleration alignment. The detailed discussion and analysis are provided in the appendix.

## 4 Experiments

### 4.1 Implementation details

**Experimental settings**. If not specified, the generated images are based on Flux.1.0-dev [28], an advanced open-sourced Rectified Flow model based on DiT architecture. The sampling steps are set as 30, the noise-adding ratio $\tau$ in initialization alignment is set as [0.6, 0.3, 0.3] for 1K → 2K → 3K → 4K cascade generation, and the normalized cutoff frequency is set as $D = 0.4$. The guidance weights in direction/acceleration alignment are set as $\alpha_t = \beta_t = t/\tau$ for gradually weakening control. All experiments are conducted on a single NVIDIA A100 GPU.

**Baselines.** The compared methods encompass DemoFusion [12], DiffuseHigh [26], I-Max [13], and an image super-resolution method, BSRGAN [58]. For a fair comparison, the training-free methods are tested on the same Flux model according to their official implementations.

**Evaluation**. We collect 1K high-quality captions across various scenarios for diverse image generation. The CLIP [43] score is used to assess the prompt-following capability, Frechet Inception Distance [18] (FID) and Inception Score [47] (IS) are reported to measure image quality, in which FID is calculated between generated images and 10K real high-quality images (with at least $1024 \times 1024$ resolution) sourced from LAION-High-Resolution [48]. Furthermore, the patch-version FID$_{\text{patch}}$ and IS$_{\text{patch}}$ are calculated based on local image patches to quantify the quality of synthesized details.

Table 1: **Quantitative comparison with other baselines.** The best result is highlighted in **bold**, while the second-best result is underlined. * indicates methods adapted from U-Net architecture.

| Resolution (height × width) | Method | FID ↓ | FID$_{\text{patch}}$ ↓ | IS ↑ | IS$_{\text{patch}}$ ↑ | CLIP Score ↑ |
|---|---|---|---|---|---|---|
| **2048 × 2048 (2K)** | DemoFusion* [12] | 56.07 | 51.69 | 27.23 | 13.48 | 35.05 |
| | DiffuseHigh* [26] | 61.62 | 50.25 | 26.76 | 13.10 | 34.83 |
| | I-Max [13] | 57.57 | 54.56 | **28.84** | 12.07 | 34.96 |
| | Flux + BSRGAN [58] | 60.25 | 52.06 | 25.85 | 13.39 | **35.34** |
| | **HiFlow (Ours)** | **55.39** | **47.70** | 28.67 | **13.86** | 35.32 |
| **4096 × 4096 (4K)** | DemoFusion* [12] | 56.72 | 49.48 | 21.17 | 8.49 | 35.27 |
| | DiffuseHigh* [26] | 62.01 | 50.98 | 20.60 | 8.09 | 34.98 |
| | I-Max [13] | 53.27 | 52.93 | 22.21 | 7.65 | 35.05 |
| | Flux + BSRGAN [58] | 59.53 | 54.12 | 19.32 | 8.87 | 35.37 |
| | **HiFlow (Ours)** | **52.55** | **45.01** | 24.62 | 9.73 | **35.40** |

### 4.2 Comparison to state-of-the-art methods

**Qualitative comparison**. As shown in Fig. 5, while DemoFusion and DiffuseHigh are capable of preserving the overall image structure, they often synthesize low-fidelity details and suffer from reduced contrast, primarily due to insufficient guidance in high-resolution detail synthesis. I-Max sometimes produces blurred results because it uses upsampled low-resolution images as the ideal projected flow endpoints, leading to overly limited detail creation. Although BSRGAN, an image super-resolution method, enhances image clarity to some extent, it struggles to synthesize finer details. In comparison, the proposed HiFlow consistently yields aesthetically pleasing and semantically coherent outcomes. More visual results are provided in the appendix. Additionally, it is observed that HiFlow demonstrates the potential for correcting local artifacts in low-resolution images in the subsequent high-resolution generation process, as illustrated in Fig. 6.

**Quantitative evaluation**. The quantitative results are presented in Tab. 1. It is observed that the proposed HiFlow achieves competitive performance in terms of both image quality (FID, FID$_{\text{patch}}$, IS and IS$_{\text{patch}}$) and image-text alignment (CLIP score) under different resolutions, validating its stable and superior performance in high-resolution image generation. Moreover, as presented in Tab. 2, HiFlow surpasses all its training-free competitors in inference speed, offering higher efficiency.

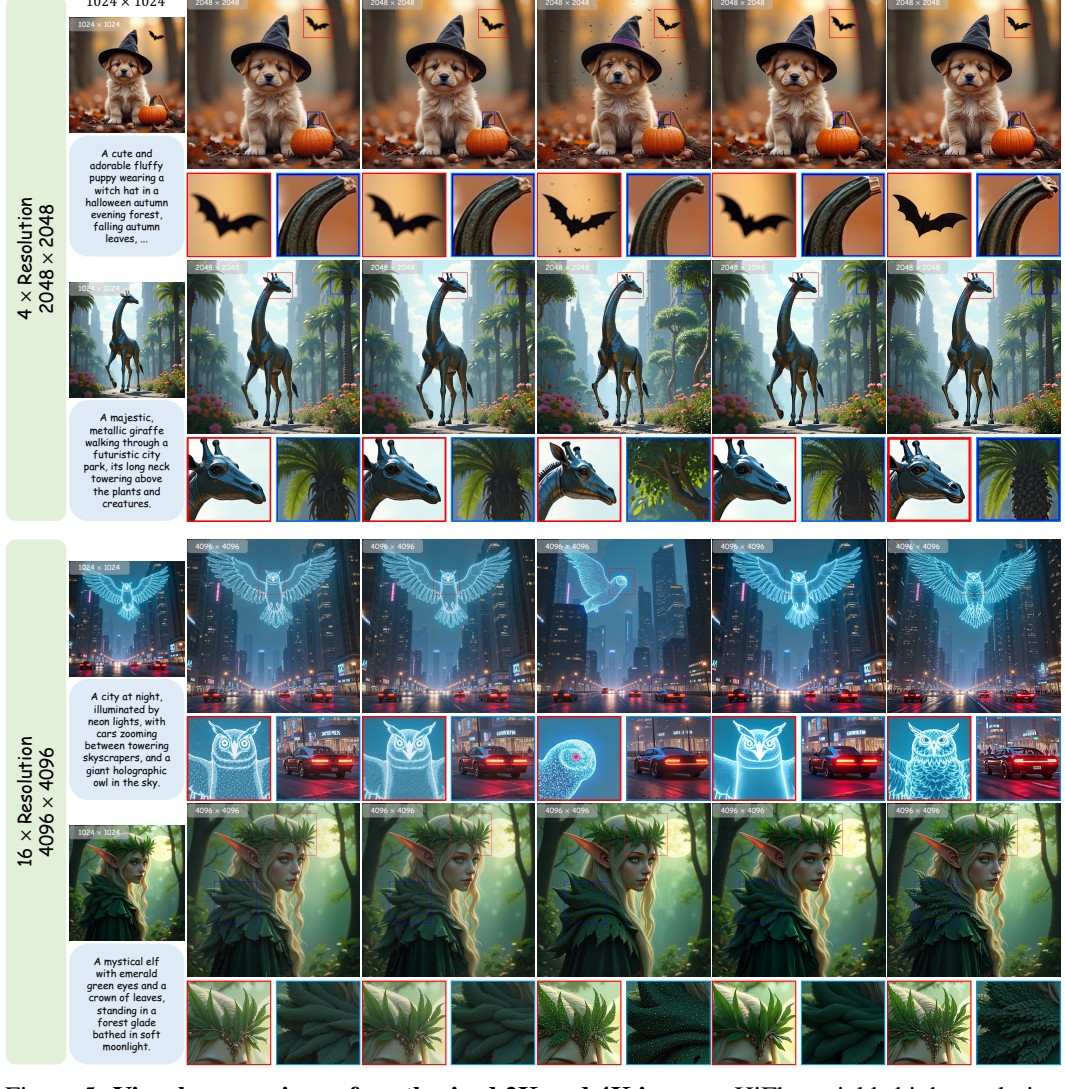

Figure 5: **Visual comparison of synthesized 2K and 4K images.** HiFlow yields high-resolution images characterized by high-fidelity details and coherent structure. Best viewed zoomed in.

## 4.3 Ablation study

HiFlow performs guided high-resolution generation by aligning its flow with the reference flow in three aspects: initialization ($A_i$), direction ($A_d$), and acceleration ($A_a$). As can be observed in Fig. 7, initialization alignment helps avoid semantic incorrectness by facilitating low-frequency consistency with low-resolution images. Furthermore, direction alignment enhances structure preservation in the generation process by suppressing the updating in the low-frequency component. Moreover, acceleration alignment contributes to high-fidelity detail synthesizing, eliminating the production of repetitive patterns in the garden, as shown in the realistic woman's face and the appearance of her clothing. Quantitative results of the ablation study are shown in Tab. 3.

## 4.4 Applications

**Application on customization T2I models**. Customization T2I models, which allow tailored generation to match users' specific requirements, have recently garnered increasing attention. As illustrated in Fig. 8 (a)-(b), the proposed HiFlow is compatible with various customization modules (LoRA [22] and ControlNet [59]), thus advancing personalized high-resolution image generation.

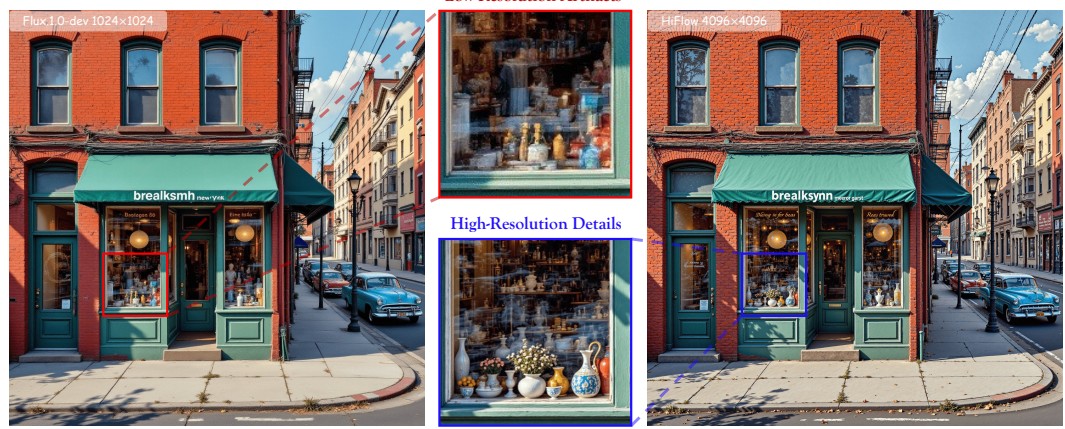

Figure 6: **HiFlow corrects local artifacts in the low-resolution result.**

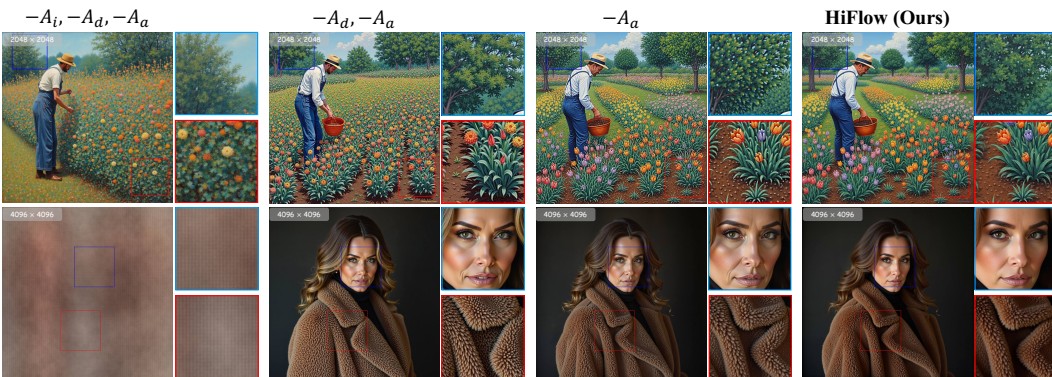

Figure 7: **Ablation experiments in 2K and 4K resolution.** Best viewed zoomed in.

**Application on quantized T2I models**. Given the substantial rise in computational complexity introduced by advanced T2I models, model quantization techniques [8, 31, 61, 62] have been widely explored to decrease computing resource requirements. As shown in Fig. 8 (c), the training-free and model-agnostic attributes of HiFlow allow it to be directly employed with the 4-bit version of Flux (quantized by SVDQuant [31]), thereby substantially accelerating high-resolution image generation.

**Application on U-Net based T2I models**. U-Net-based T2I models, such as SD1.5 [45] and SDXL [41], constitute pivotal parts of T2I diffusion models and have been extensively developed by the community. As depicted in Fig. 8 (d), the integration of HiFlow and SDXL [41] enables the synthesis of realistic high-resolution images, demonstrating its broad applications.

Table 2: **Comparison in latency.**

| Resolution | Method | Latency $(sec.)$ ↓ |
|---|---|---|
| $2048^2$ | DemoFusion* [12] | 106 |
| | DiffuseHigh* [26] | 59 |
| | I-Max [13] | 94 |
| | **HiFlow (Ours)** | **56** |
| $4096^2$ | DemoFusion* [12] | 972 |
| | DiffuseHigh* [26] | 533 |
| | I-Max [13] | 735 |
| | **HiFlow (Ours)** | **379** |

Table 3: **Quantitative results of ablation study.**

| Resolution | Method | FID ↓ | FID$_{patch}$ ↓ | IS ↑ | IS$_{patch}$ ↑ | CLIP ↑ |
|---|---|---|---|---|---|---|
| $2048^2$ | $-A_a, -A_d, -A_i$ | 67.87 | 71.79 | 23.47 | 9.16 | 33.81 |
| | $-A_a, -A_d$ | 58.26 | 54.22 | 27.46 | 12.58 | 34.47 |
| | $-A_a$ | 58.40 | 50.38 | 28.92 | 13.14 | 34.90 |
| | **HiFlow (Ours)** | **55.39** | **47.70** | 28.67 | **13.86** | **35.32** |
| $4096^2$ | $-A_a, -A_d, -A_i$ | 234.38 | 198.21 | 9.33 | 3.31 | 11.42 |
| | $-A_a, -A_d$ | 56.20 | 48.30 | 19.85 | 6.51 | 33.64 |
| | $-A_a$ | 55.36 | 51.79 | 22.78 | 8.77 | **35.44** |
| | **HiFlow (Ours)** | **52.55** | **45.01** | 24.62 | 9.73 | 35.40 |

## 5 Conclusion

We present HiFlow, a tuning-free and model-agnostic framework enabling pre-trained Rectified Flow models to generate high-resolution images with high fidelity and rich details. HiFlow involves a novel cascade generation paradigm: (i) a virtual reference flow is constructed based on the step-wise predicted clean samples of the low-resolution sampling trajectory; (ii) the high-resolution flow is

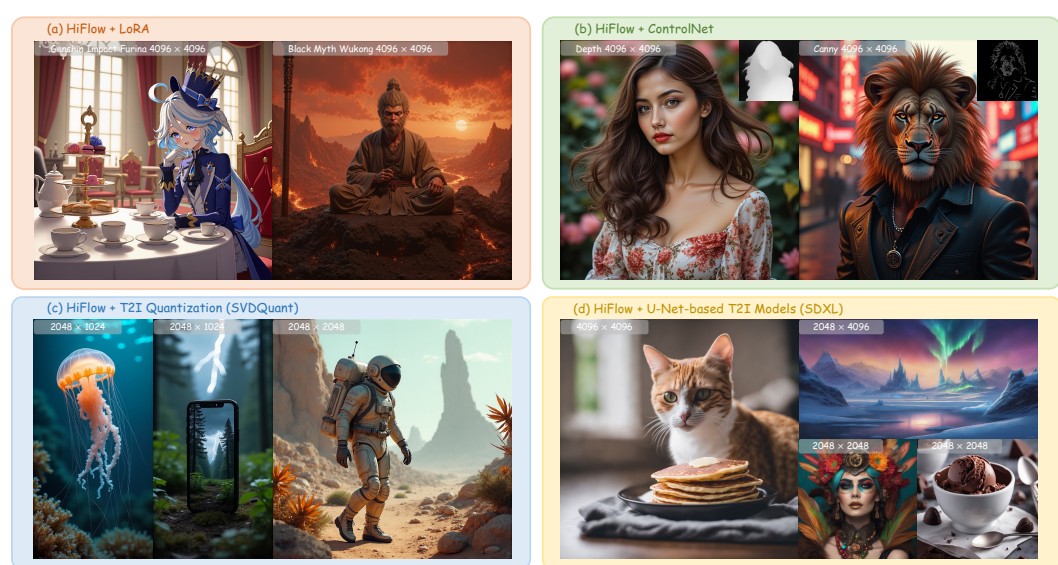

Figure 8: **Versatile applications of HiFlow.**

aligned with the reference flow via flow-aligned guidance, which encompasses three aspects: initialization alignment for low-frequency consistency, direction alignment for structure preservation, and acceleration alignment for detail fidelity. Extensive experiments demonstrate that HiFlow surpasses state-of-the-art methods and highlight its broad applicability across different model architectures.

# 6 Acknowledgement

This project is funded in part by Shanghai Artificial lntelligence Laboratory, Shanghai Innovation Institute, the National Key R&D Program of China (2022ZD0160201), the Centre for Perceptual and Interactive Intelligence (CPII) Ltd under the Innovation and Technology Commission (ITC)'s InnoHK. Dahua Lin is a PI of CPII under the InnoHK.

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

# Appendix

In the appendix, we present additional implementation details (Section A), additional qualitative results (Section B), text prompts for image generation in both the main paper and appendix (Section C), more discussion on acceleration alignment (Section D), further derivations regarding $X_{1\leftarrow t}$ and $X_{0\leftarrow t}$ (Section E), pseudo code of flow-aligned guidance (Section F), the limitations of our method (Section G) as well as its broader impacts (Section H), as a supplement to the main paper.

## A    Additional implementation details

To enhance the Rectified Flow model's stability when generalizing to extrapolated resolutions, we adopt inference techniques suggested by previous works [39, 24, 13] in the high-resolution generation process, including NTK-aware scaled RoPE [39], balancing the entropy shift of self-attention [24] and balancing the image/text sequence length ratio for MMDiT [13].

## B    Additional qualitative results

We present more visual results of HiFlow integrated with various LoRA models at $4096 \times 4096$ resolution in Fig. H.4, Fig. H.5, Fig. H.6, Fig. H.7, Fig. H.8 and Fig. H.9, along with generated results using the same prompts and different random seeds $(0/1/2)$ at $4096 \times 4096$ resolution in Fig. H.10. Moreover, additional results of HiFlow on SVDQuant [31] and SDXL [41] are shown in Fig. H.11.

## C    Text prompts

Text prompts used to generate images in this paper are provided in Tab. H.1, Tab. H.2 and Tab. H.3.

## D    Discussion on acceleration alignment

In this section, we provide a deeper analysis on acceleration alignment. Given that Rectified Flow models are based on the premise that the noisy latent variable $X_t$ transitions linearly between between the noise distribution $\pi_1$ and the clean image distribution $\pi_0$, it is logical to introduce an acceleration term drawn from physics to account for non-linear dynamics during the denoising process:

$$v_t = \frac{dX_t}{dt}, a_t = \frac{dv_t}{dt} = \frac{d^2 X_t}{dt^2}, \tag{14}$$

where $v_t$ is the denoising direction predicted by the model (the first derivative of $X_t$ with respect to $t$), and $a_t$ denotes the acceleration component (the first derivative of $v_t$ with respect to $t$, the second derivative of $X_t$ with respect to $t$). The above definition is consistent with previous works [7] that introduced acceleration in image editing tasks. To explore the practical implication of $a_t$, Eq. 4 is substituted into the definition of acceleration in Eq. 14:

$$a_t = \frac{dv_t}{dt} = \frac{v_{t_{i-1}} - v_{t_i}}{t_{i-1} - t_i} = \frac{\frac{1}{t_{i-1}}(X_{t_{i-1}} - X_{0\leftarrow t_{i-1}}) - \frac{1}{t_i}(X_{t_i} - X_{0\leftarrow t_i})}{t_{i-1} - t_i}. \tag{15}$$

Furthermore, based on Eq. 4 and Eq. 5, we are able to express $X_{t_{i-1}}$ in terms of $X_{t_i}$, $X_{0\leftarrow t_i}$ and $X_{0\leftarrow t_{i-1}}$ through the following relation:

$$X_{t_{i-1}} = X_{t_i} + v_{t_i}(t_{i-1} - t_i) = X_{t_i} + \frac{X_{t_i} - X_{0\leftarrow t_i}}{t_i}(t_{i-1} - t_i) = X_{0\leftarrow t_i} + \frac{t_{i-1}}{t_i}(X_{t_i} - X_{0\leftarrow t_i}). \tag{16}$$

By plugging Eq. 16 into Eq. 15, $X_{t_{i-1}}$ is eliminated and the expression for $a_t$ can be simplified as:

$$\begin{aligned} a_t &= \frac{\frac{1}{t_{i-1}}(X_{0\leftarrow t_i} + \frac{t_{i-1}}{t_i}(X_{t_i} - X_{0\leftarrow t_i}) - X_{0\leftarrow t_{i-1}}) - \frac{1}{t_i}(X_{t_i} - X_{0\leftarrow t_i})}{t_{i-1} - t_i} \\ &= -\frac{1}{t_i}\frac{X_{0\leftarrow t_i-1} - X_{0\leftarrow t_i}}{t_{i-1} - t_i}, \\ &= -\frac{1}{t_i}\frac{dX_{0\leftarrow t}}{dt}. \end{aligned} \tag{17}$$

From Eq. 17, it is concluded that the acceleration term essentially depicts the first order derivative of the predicted clean sample $X_{0\leftarrow t}$ with respect to $t$, multiplied by a time-dependent factor $-\frac{1}{t}$. Therefore, the acceleration in the denoising process of Rectified Flow models primarily captures the variance in image details, including what details should be synthesized at each timestep and how these details should be synthesized for better fidelity, especially in the late denoising stage when the model focuses on the generation of fine image details. Fig. D.1 provides a visualization of the acceleration across different timesteps, and the results are consistent with our analysis.

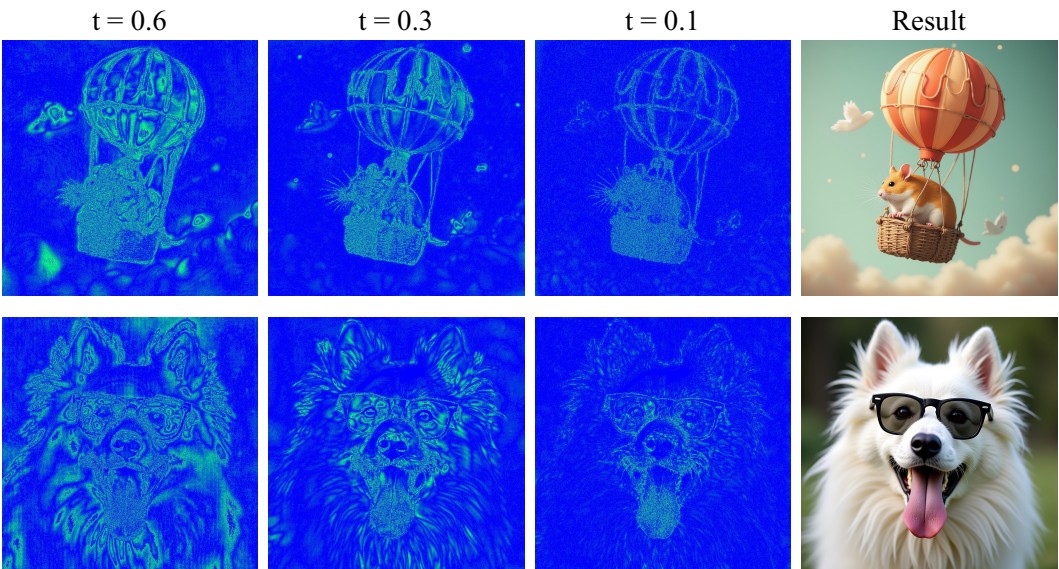

Figure D.1: **Visualization of acceleration at different timesteps.**

The above discussion indicates that acceleration characterizes the model's ordering, rhythm, and preferences in synthesizing image details, while the reference flow reflects how this information, learned at the training resolution, manifests in the high-resolution space. To this end, the acceleration alignment in Eq. 11 is introduced to guide the acceleration in high-resolution generation in the form of classifier-free guidance [20], for better detail fidelity.

## E    Further derivations regarding $X_{1\leftarrow t}$ and $X_{0\leftarrow t}$

In this paper, $X_{0\leftarrow t}$ denotes the predicted/estimated clean component of the current noisy latent $X_t$, while $X_{1\leftarrow t}$ represents its corresponding noise component at timestep $t$. For rectified flow models, the model output $v_\theta(X_t, t, c)$ stands for the flow vector/velocity (we use $v_t$ to denote $v_\theta(X_t, t, c)$ for simplicity). Given the model output $v_t$ and the current noisy latent $X_t$ at timestep $t$, the estimated clean sample $X_{0\leftarrow t}$ and its corresponding estimated noise $X_{1\leftarrow t}$ can be obtained by:

$$X_{0\leftarrow t} = X_t - v_t t, \tag{18}$$

$$X_{1\leftarrow t} = X_t + v_t(1 - t). \tag{19}$$

A visualization example of $X_{0\leftarrow t}$, $X_{1\leftarrow t}$ and $X_t$ during the denoising process is presented in Fig. E.2.

## F    Pseudo code of flow-aligned guidance

In this section, we present the pseudo code for each component of the proposed flow-aligned guidance, as detailed in Algorithm 1.

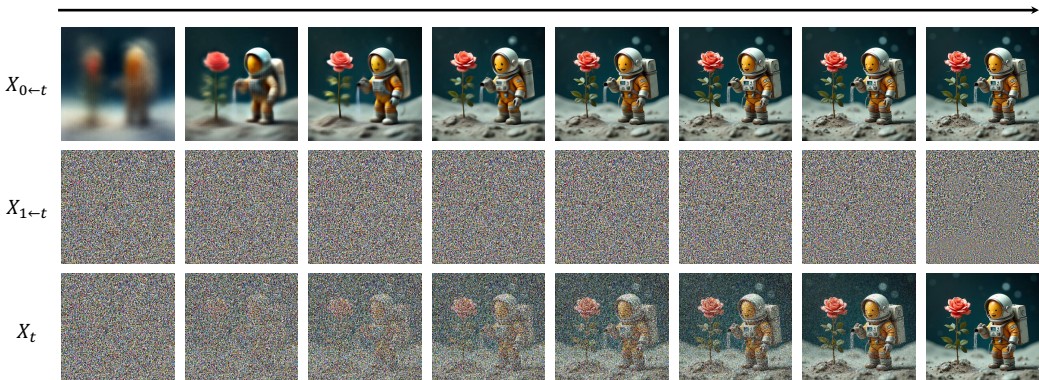

Figure E.2: **Visualization of** $X_{0\leftarrow t}$, $X_{1\leftarrow t}$ **and** $X_t$ **at different timesteps.**

---

**Algorithm 1** Flow-Aligned Guidance

---

```
# Initialization Alignment
def Initialization_Alignment(reference_flow, noise_adding_ratio):
    tau = noise_adding_ratio
    X_0_ref = reference_flow.predict_clean_samples[tau]
    noise = RandNoise(X_0_ref.shape)
    X_0_high = AddNoise(X_0_ref, noise, tau)
    return X_0_high

# Direction Alignment
def Direction_Alignment(high_resolution_flow, reference_flow, t, alphas):
    X_0_high = high_resolution_flow.predict_clean_samples[t]
    X_0_ref = reference_flow.predict_clean_samples[t]
    X_0_high = X_0_high + alphas[t] * (LowFreq(X_0_ref) - LowFreq(X_0_high))
    return X_0_high

# Acceleration Alignment
def Acceleration_Alignment(high_resolution_flow, reference_flow, t, betas):
    v_t_high = high_resolution_flow.flow_vectors[t]
    v_t_high_prev = high_resolution_flow.flow_vectors[t + 1]
    v_t_ref = reference_flow.flow_vectors[t]
    v_t_ref_prev = reference_flow.flow_vectors[t + 1]
    v_t_high = v_t_high + betas[t] * (v_t_ref - v_t_ref_prev - v_t_high + v_t_high_prev)
    return v_t_high

# Notice that Initialization Alignment is performed outside the sampler
def Sampler_with_Flow_Aligned_Guidance(high_resolution_flow, reference_flow, t, alphas, betas):
    v_t_high = high_resolution_flow.flow_vectors[t]
    # Direction Alignment
    high_resolution_flow.predict_clean_samples[t] = EstimateCleanSample(v_t_high)
    X_0_high = Direction_Alignment(high_resolution_flow, reference_flow, t, alphas)
    # Acceleration Alignment
    high_resolution_flow.flow_vectors[t] = FlowVector(X_0_high)
    v_t_high = Acceleration_Alignment(high_resolution_flow, reference_flow, t, betas)
    # Sample with aligned high-resolution flow
    next_noisy_latent = FlowMatchingSampler(high_resolution_flow.noisy_latents[t], v_t_high, t)
    return next_noisy_latent
```

---

# G   Limitation and discussion

Despite the advancements of HiFlow in enhancing high-resolution image generation, it faces certain constraints. As a training-free method, the generation capability of HiFlow is highly dependent on the quality of the reference flow, which provides flow-aligned guidance in each sampling step. Therefore, the structural irregularities within the reference flow might compromise the quality of the final production. As shown in Fig. G.3, the low-resolution image of a tiger exhibits a duplicated tail, and this structural anomaly is preserved in the high-resolution generation result. Fortunately, given HiFlow's training-free and model-agnostic nature, such flaws can be effectively mitigated by (i) integrating HiFlow into a stronger backbone model, (ii) using ControlNet [59] to introduce external control conditions for controllable generation (see Fig. 8 (b) in the main paper), or (iii)

applying inference-time scaling techniques [37] to search for a better low-resolution trajectory for the subsequent high-resolution generation, bypassing the need for fine-tuning or adjustment.

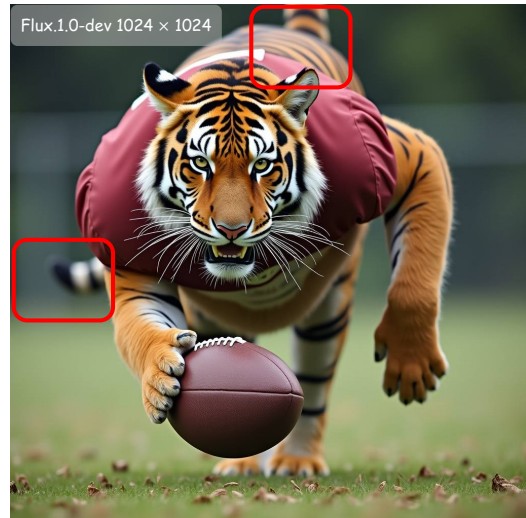 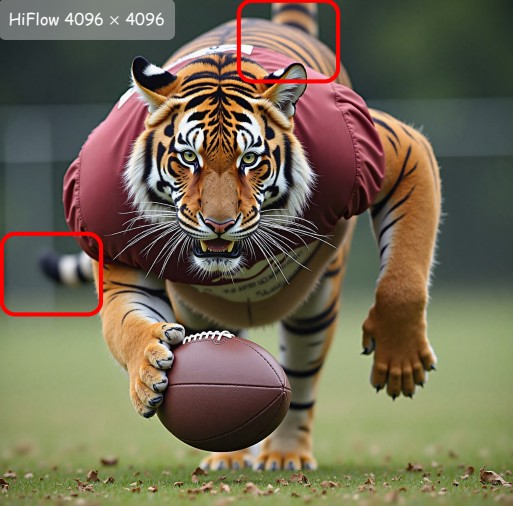

Figure G.3: **Failure case of HiFlow.**

## H  Broader impacts

The introduction of HiFlow, a novel training-free framework for high-resolution image generation, brings with it a range of societal impacts, offering significant benefits while also posing notable risks.

On the beneficial side, HiFlow's ability to produce detailed, high-resolution images directly from textual prompts without requiring model retraining opens up exciting possibilities across multiple domains. In creative industries, such as design, marketing, and digital media, professionals can rapidly prototype visuals, enhance visual storytelling, and generate production-ready assets with minimal cost and effort. The high resolution and quality of the outputs make them suitable for both conceptual and final-use purposes. In education and scientific communication, HiFlow can assist instructors and researchers in generating accurate visual aids tailored to specific topics, enriching the learning experience—especially in subjects that rely on precise imagery, like anatomy, architecture, or environmental science.

However, this powerful capability also introduces challenges. The ease and accessibility of generating photorealistic images without any fine-tuning may lead to the creation and spread of deceptive or harmful content, including synthetic news photos, manipulated evidence, or deepfake creation. Without robust safeguards, such misuse could contribute to misinformation, digital fraud, or the erosion of trust in visual media.

In conclusion, while HiFlow marks a significant milestone in the realm of high-resolution image generation, its responsible use requires the establishment of ethical standards, transparent practices, and public awareness. As the technology continues to advance, collaboration among developers, regulators, and society at large will be essential to ensure that its benefits are maximized while minimizing potential harms. By fostering accountability and encouraging the development of verification mechanisms, we are able to guide the use of HiFlow toward constructive and trustworthy applications.

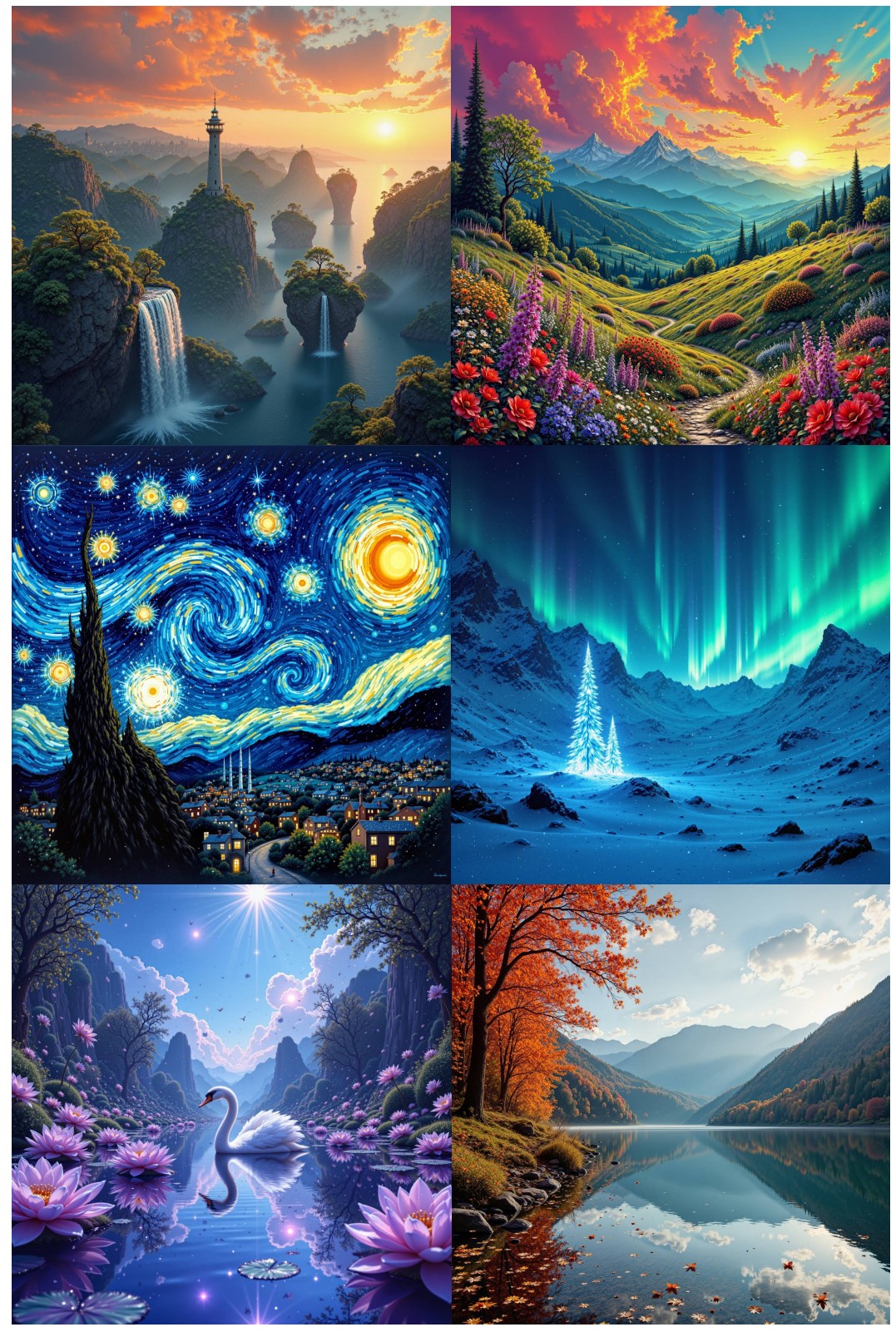

Figure H.4: **Visual results of HiFlow at** $4096 \times 4096$ **resolution.**

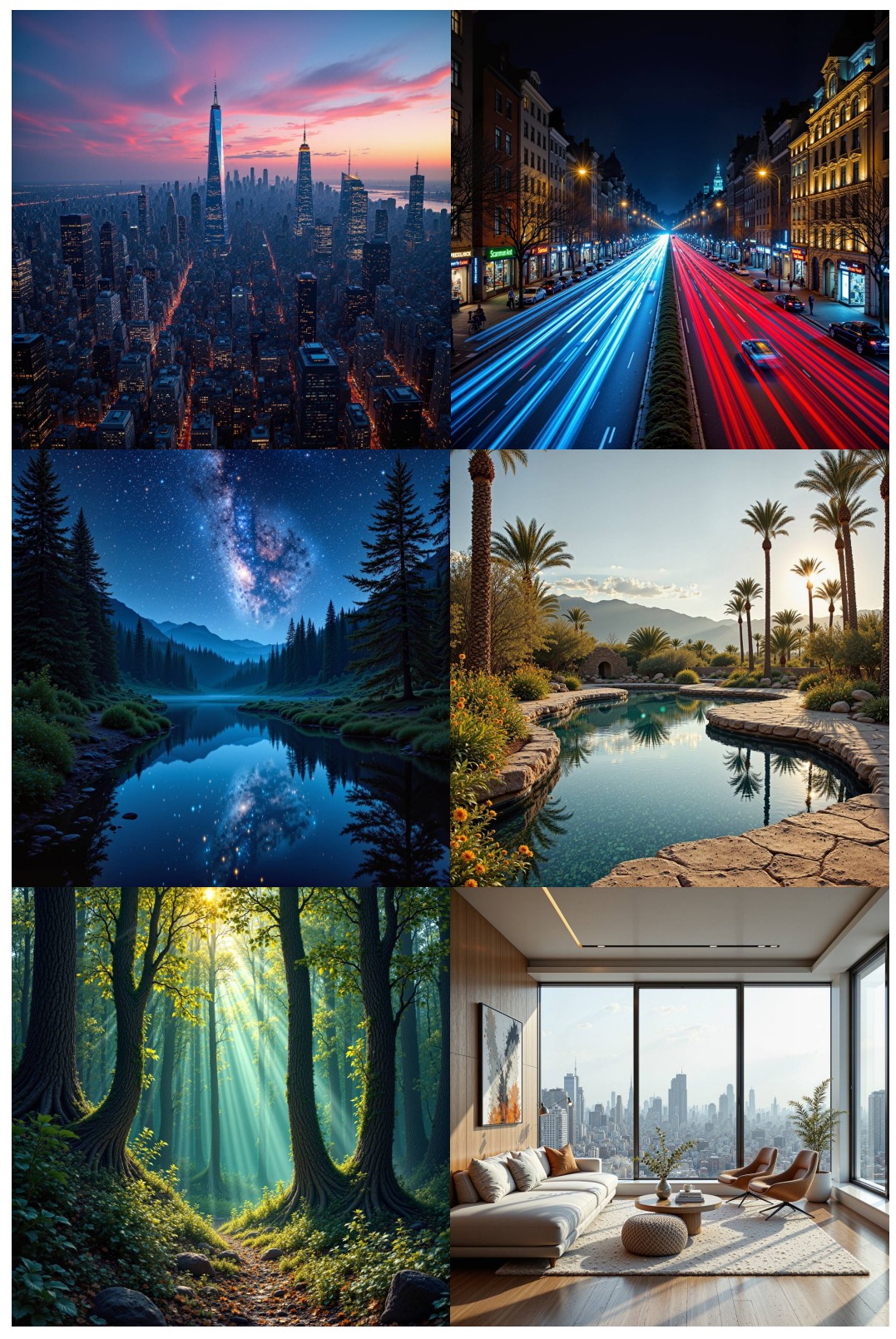

Figure H.5: **Visual results of HiFlow at** $4096 \times 4096$ **resolution.**

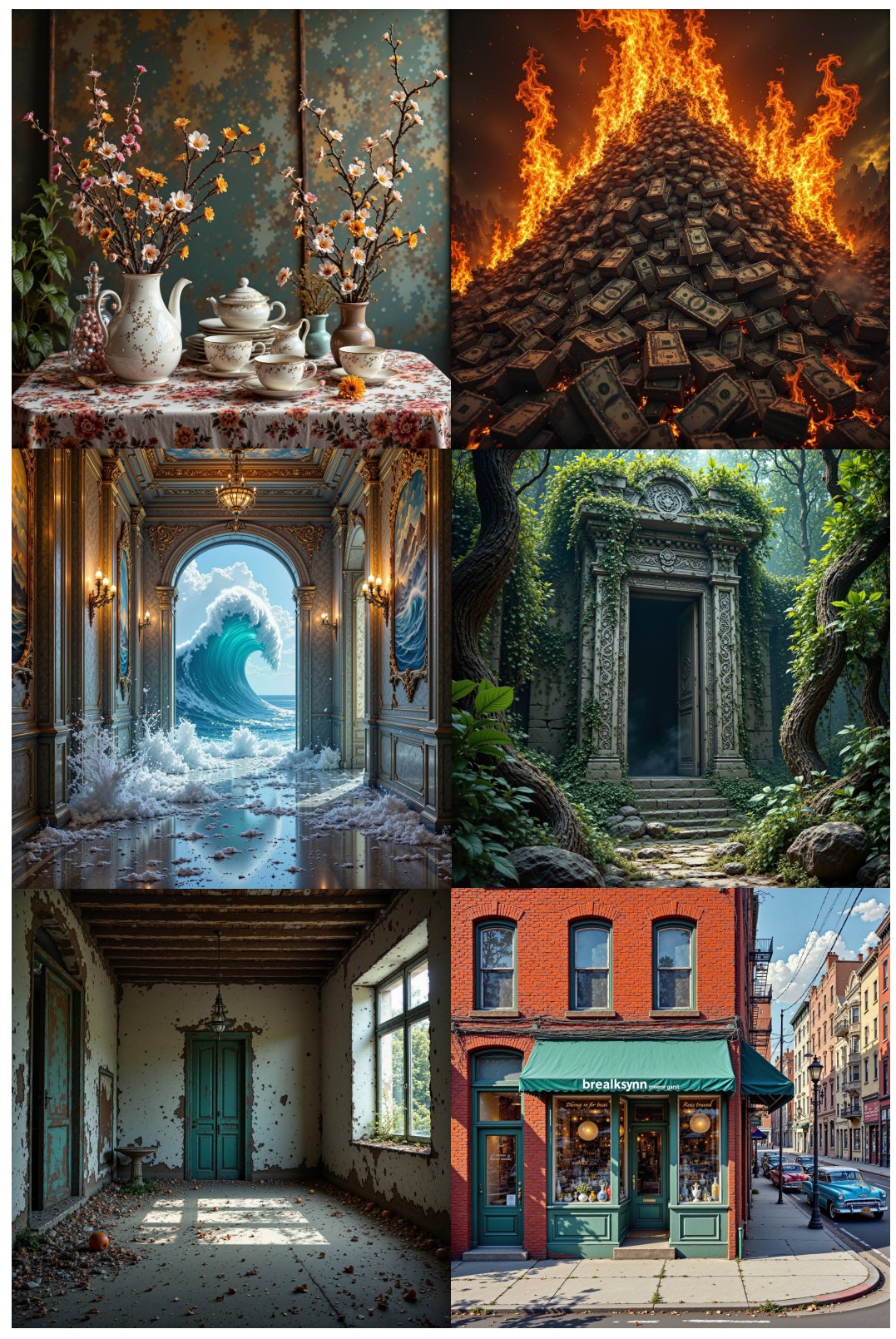

Figure H.6: **Visual results of HiFlow at** $4096 \times 4096$ **resolution.**

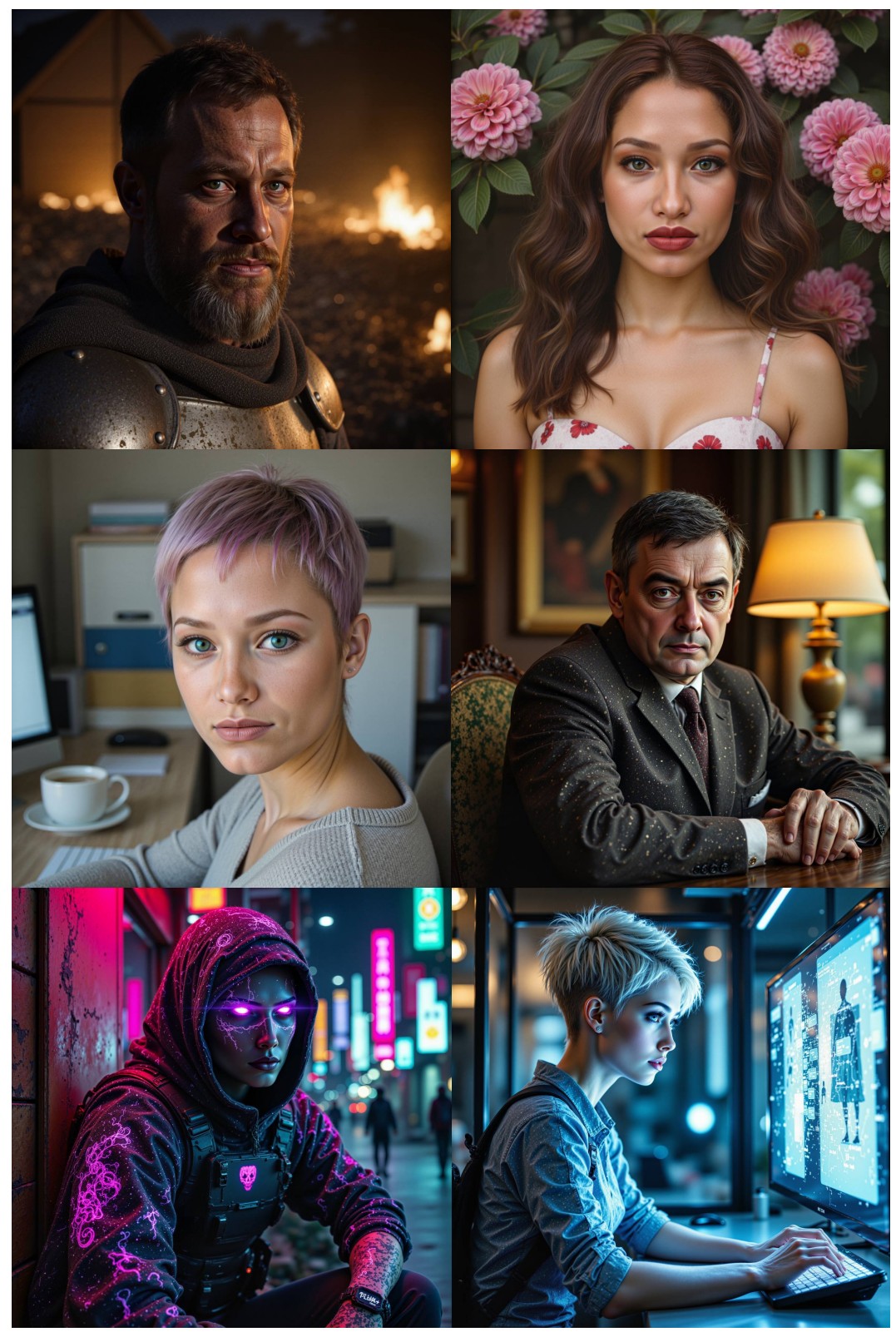

Figure H.7: **Visual results of HiFlow at** $4096 \times 4096$ **resolution.**

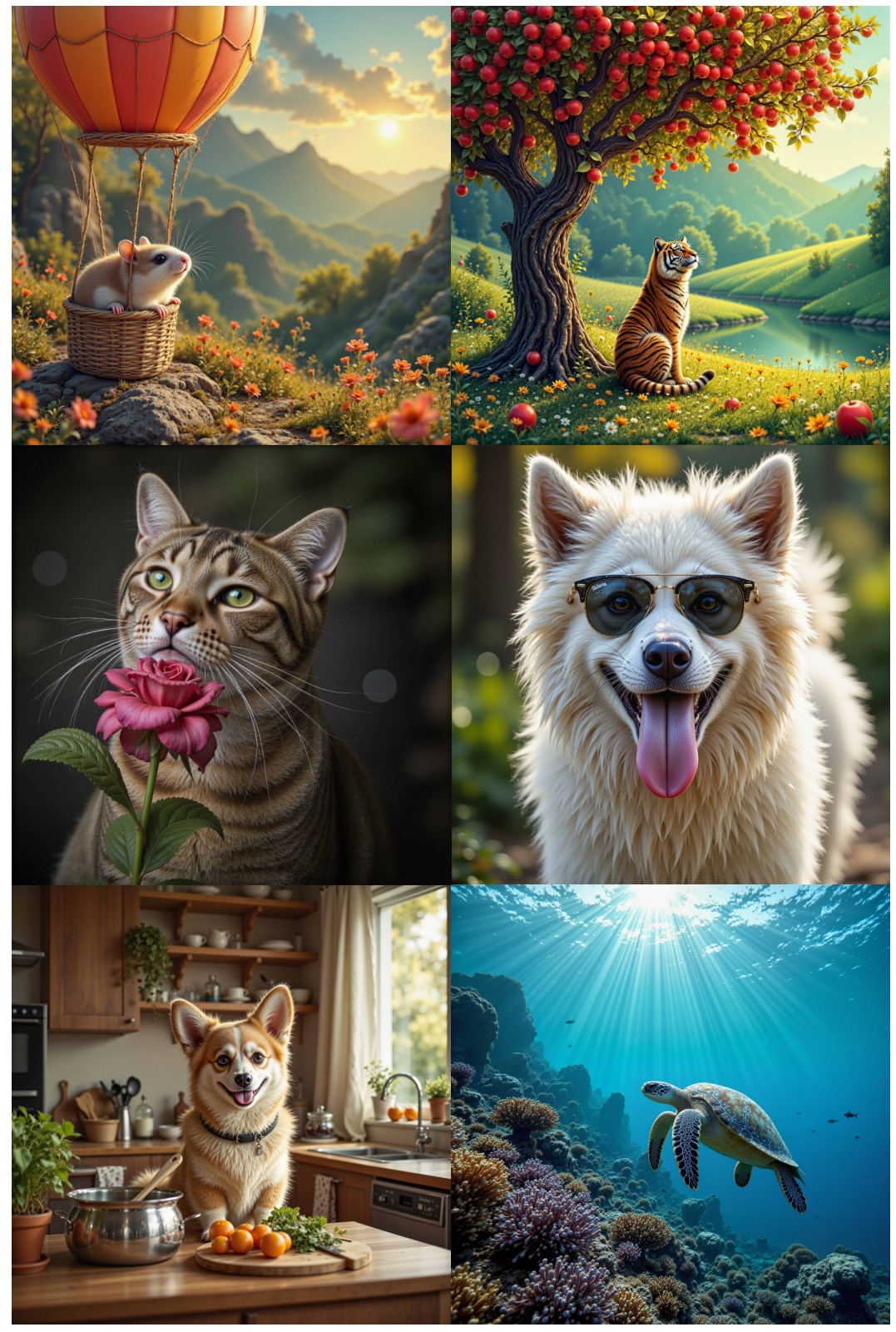

Figure H.8: **Visual results of HiFlow at** $4096 \times 4096$ **resolution.**

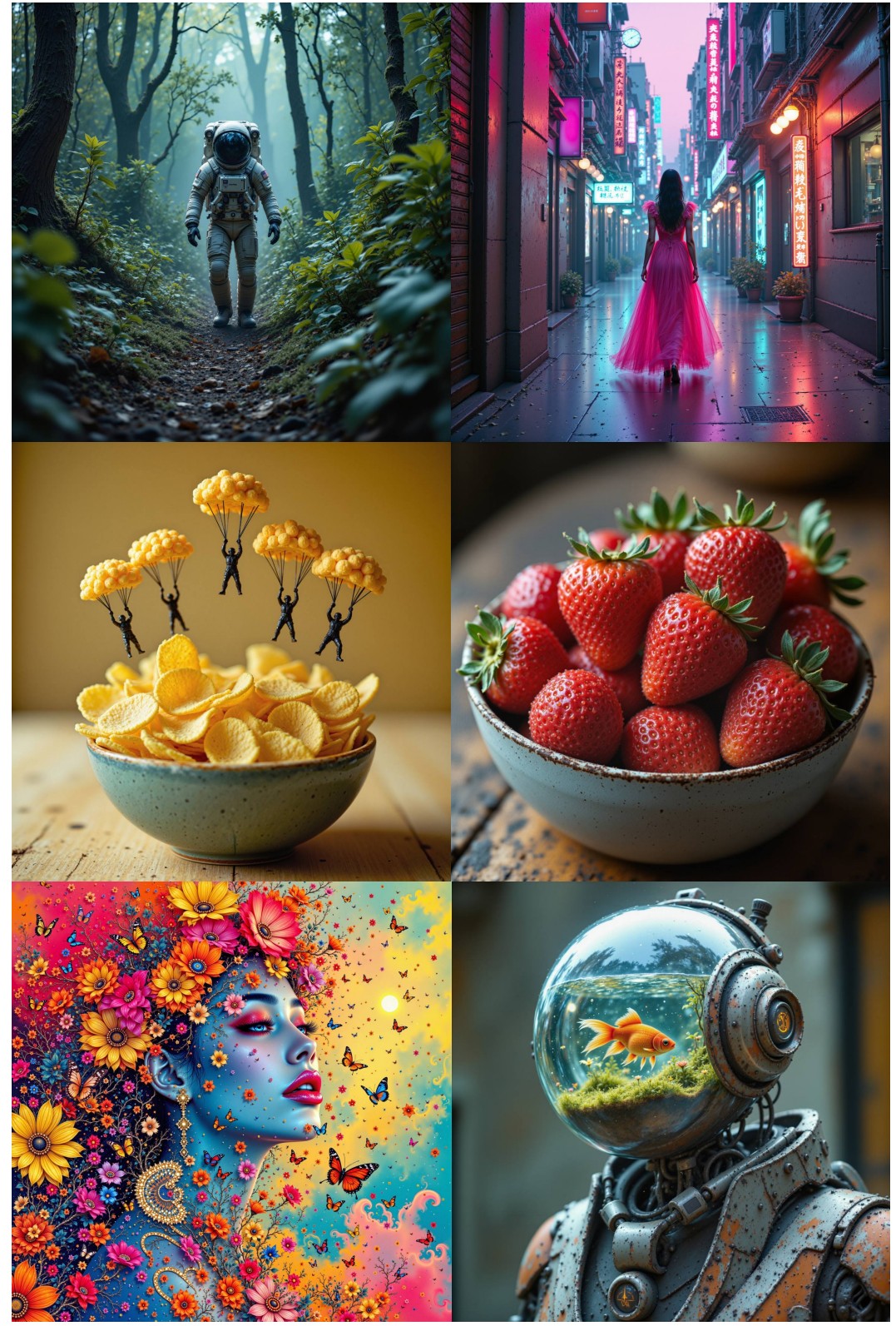

Figure H.9: **Visual results of HiFlow at** $4096 \times 4096$ **resolution.**

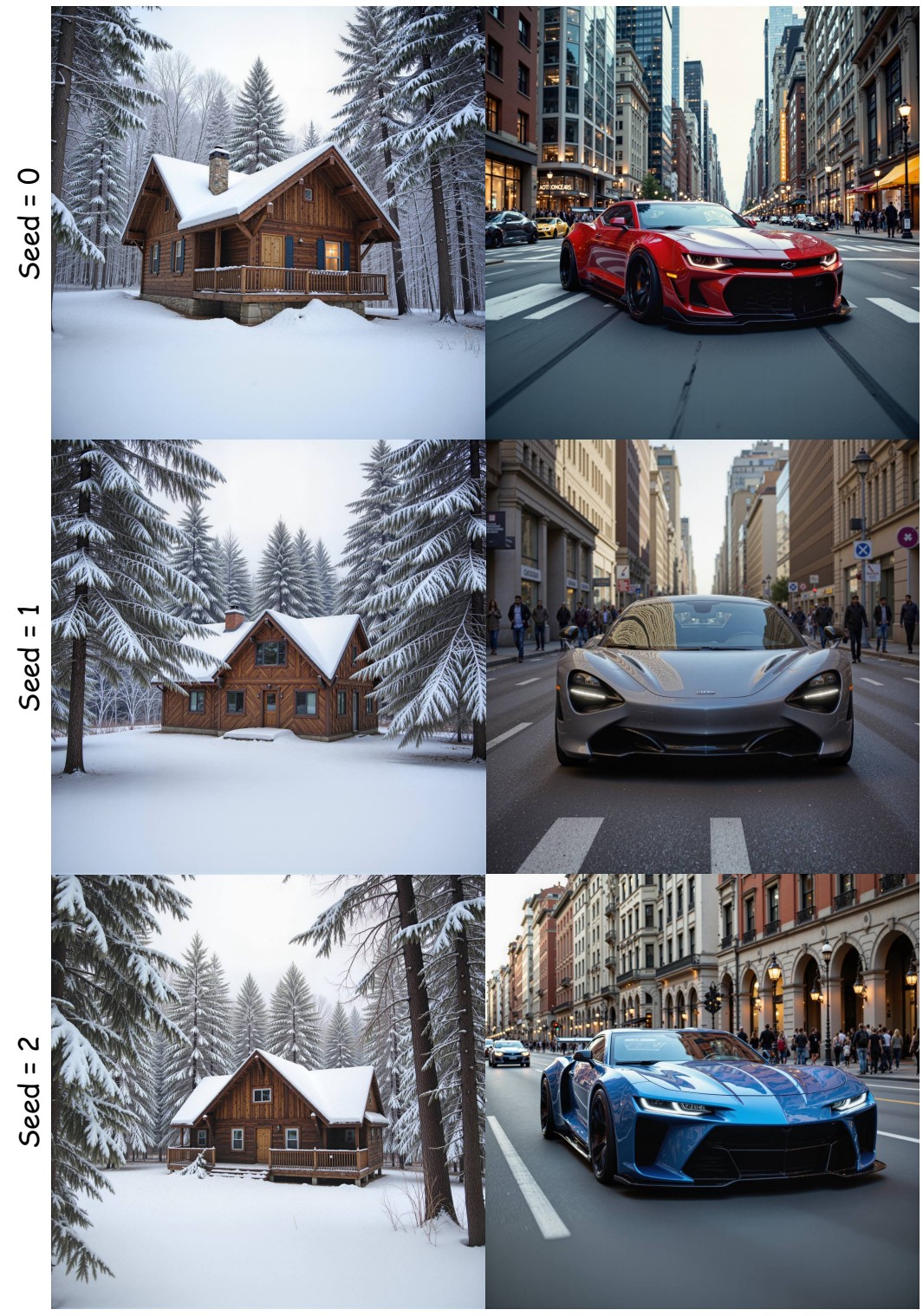

Figure H.10: **Generated results using same prompts and different seeds at** $4096 \times 4096$ **resolution.**

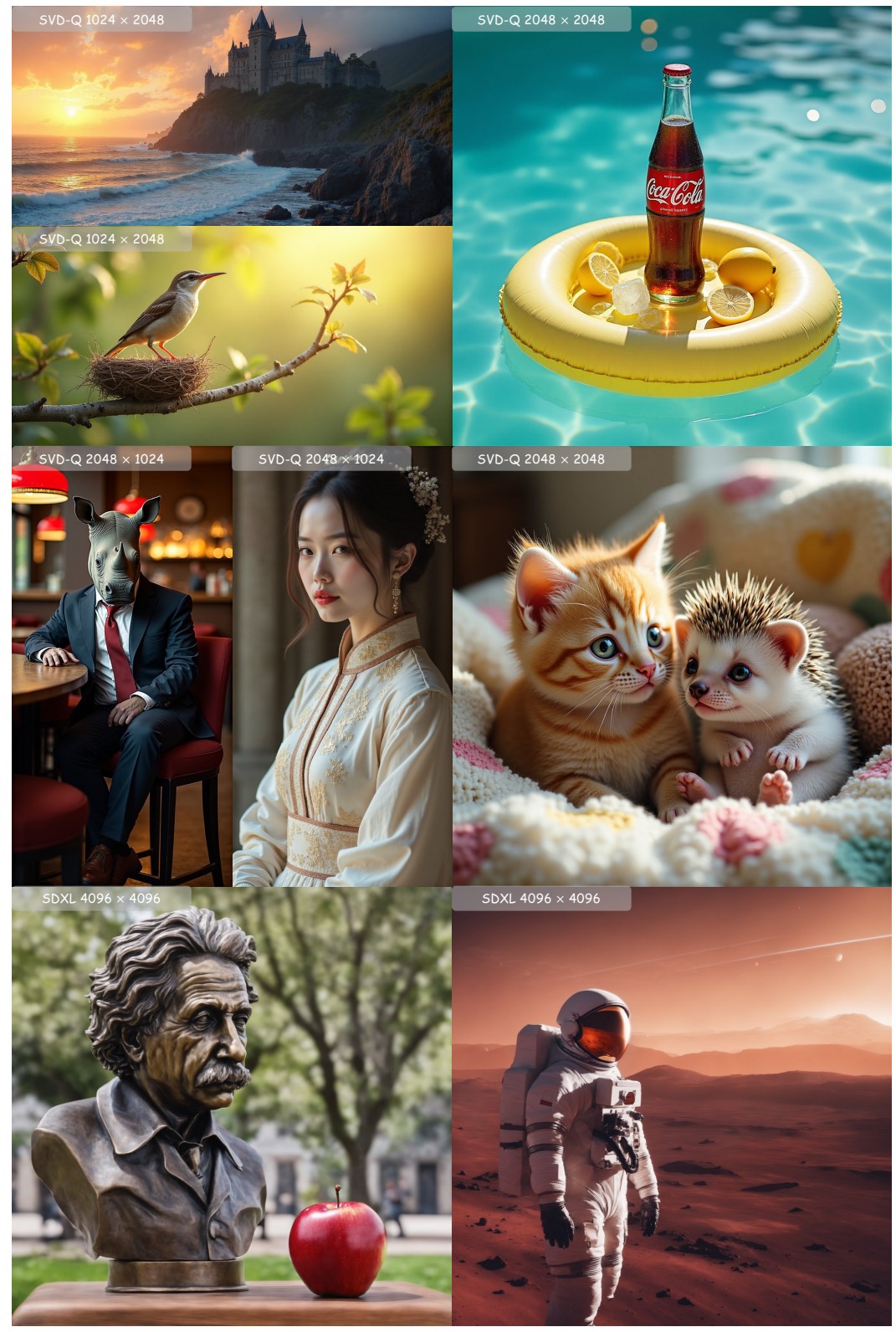

Figure H.11: **More visual results of HiFlow on SVDQuant and SDXL.**

Table H.1: The image generation prompts for each figure are listed sequentially, following the order from left to right and top to bottom. (Table 1/3)

| Figure | Text Prompt |
| --- | --- |
| Figure 1 | Knight holding kitten in flower garden: A knight in full plate armor stands amidst blooming flowers, gently cradling a tabby kitten in their gauntleted hands. Sunlight filters through the foliage, creating a warm, dappled light. The kitten looks up at the knight with a curious expression. Focus on the contrast between the hard armor and the soft fur. Photorealistic style with a touch of fantasy. |
| | The image depicts a detailed and dynamic illustration of a Gundam, specifically Gundam Barbatos from the anime Mobile Suit Gundam: Iron-Blooded Orphans. This powerful mecha is portrayed in a close-up action shot, emphasizing its imposing and battle-worn appearance. Gundam Barbatos features a distinct design with a white and blue color scheme, accented by gold and red details. The mecha's head is particularly striking, with a prominent yellow V-fin crest that is characteristic of many Gundam designs. Its eyes glow a vibrant green, adding to the intensity and life-like presence of the machine. The head unit also has a faceplate that suggests a fierce and determined expression, fitting for a battle-hardened warrior. The armor plating on B4RB4T0S appears worn and weathered, with visible scratches and damage, indicating that it has seen many battles. The chest area is reinforced with blue armor, and the overall structure of the Gundam is muscular and robust, reflecting its strength and combat capabilities. In the background, there are blurred elements that suggest a chaotic battlefield, with dark, smoky skies and hints of fire or explosions, adding to the dramatic atmosphere of the scene. The lighting and shading in the image are expertly done, creating a sense of depth and realism, while also highlighting the metallic surfaces of the Gundam. Overall, the image captures the essence of Gundam Barbatos as a powerful and relentless war machine, ready to engage in fierce combat. The detailed rendering and dynamic composition emphasize the Gundam's role as a central figure in the struggle depicted in Iron-Blooded Orphans. |
| | A breathtaking mountain landscape featuring towering peaks with snow caps under a serene blue sky. The scene captures the early morning light, casting soft shadows and illuminating the majestic Tetons. A tranquil river runs through, reflecting the mountains and surrounding evergreen trees, creating a mirror-like effect. The composition emphasizes depth, framing the mountains with a cluster of pines on the left and gentle frost covering the ground. The cool color palette evokes a sense of calm and tranquility, enhanced by the crisp air and delicate mist lingering over the water. The scene embodies a peaceful wilderness atmosphere, reminiscent of fine art photography. |
| | A mouthwatering photograph of a well-plated gourmet burger and French fries, with a glass of cola with ice cubes in the background. |
| | Portrait of a bear as a roman general in a roman city-state, with a helmet, decorative, fantasy environment, detailed, sharp, clear, 8k. |
| | A whimsical village scene is nestled within an enormous teacup, where winding cobblestone streets and quaint cottages create a surreal microcosm reminiscent of dreamlike landscapes. From a high vantage point, the viewer looks down on the diminutive inhabitants going about their day-to-day activities, contrasting the juxtaposition of innocence and chaos in this fantastical setting. Soft, ethereal lighting envelops the scene, creating an atmosphere of tranquility as ordinary life intertwines with elements of fantasy and absurdity to weave a captivating visual narrative that invites the audience into an extraordinary world where the commonplace becomes surreal. |
| | Super detailed, half-body portrait of a beautiful young girl with flawless skin and golden hair, wearing an elegant ball gown. Posed in an ancient and opulent castle hall with ornate decorations and grand architecture in the background. Bright, soft light illuminates her face. Clear, perfect, and detailed face with brilliant blue eyes, full red lips, a friendly smile, pale skin with freckles, and vivid colors. |
| | A fluffy, round-faced British Shorthair cat with big expressive eyes, sitting calmly on a wooden desk in a cozy study room. The background features a bookshelf filled with vintage books and decorative objects. Soft, natural lighting enhances the cozy, nostalgic atmosphere. |
| | A blood-red Ferrari SF90 parked on a rain-soaked city street at night, reflecting neon lights from nearby buildings. The wet pavement glistens, and the car's smooth curves are highlighted by the ambient glow of the urban environment. |
| | A professional photograph of a quirky squirrel with a fiery red mohawk, energetically playing a whimsical drum set in a vibrant autumn forest. The drum kit is entirely imagined, with bass drums made from hollowed, oversized chestnuts that have been polished to a gleaming finish. The snare drum is crafted from a woven basket of intertwined twigs and leaves, producing a crisp, earthy sound. Cymbals are formed from large, dry maple leaves, their edges curled and textured, hanging on thin branch stands. Smaller acorns serve as tom-toms, and the hi-hat is composed of overlapping oak leaves, delicately stacked on a twig frame. Every element of the drum set is organic, blending seamlessly into the forest floor, which is blanketed in a rich layer of autumn leaves, as the squirrel's paws expertly tap the instruments, filling the scene with the imagined rhythm of nature. |
| | Wukong, wearing the armor of the Monkey King, holding his weapon with a serious expression, inside an abandoned traditional Chinese temple. |
| | A close-up of a blooming peony, with layers of soft, pink petals, a delicate fragrance, and dewdrops glistening in the early morning light. |
| | A futuristic robotic unicorn with a chrome-plated horn and neon-glowing hooves, shredding a halfpipe in a post-apocalyptic skatepark. Sparks fly from its hooves, and graffiti-covered ruins in the background in electric blue spray paint. Dynamic motion blur, ultra-realistic metallic textures, vibrant cyberpunk colors, Mad Max meets Lisa Frank aesthetic. |
| Figure 2 | A tiny astronaut hatching from an egg on the moon. |
| | A lighthouse standing tall against crashing waves. |
| | Teddy bear walking down 5th Avenue, beautiful sunset, close up, high definition, 4k. |

Table H.2: The image generation prompts for each figure are listed sequentially, following the order from left to right and top to bottom. (Table 2/3)

| Figure | Text Prompt |
|---|---|
| Figure 3 | A pair of navy blue Converse shoes displayed on the table. |
| | A highly detailed oil painting of tiger. |
| | A woman in pink T-shirt. |
| | Small cottage near the lake, summer. |
| Figure 5 | A cute and adorable fluffy puppy wearing a witch hat in a halloween autumn evening forest, falling autumn leaves, brown acorns on the ground, halloween pumpkins spiderwebs, bats, a witch's broom. |
| | A majestic, metallic giraffe walking through a futuristic city park, its long neck towering above the plants and creatures. |
| | A city at night, illuminated by neon lights, with cars zooming between towering skyscrapers, and a giant holographic owl in the sky. |
| | A mystical elf with emerald green eyes and a crown of leaves, standing in a forest glade bathed in soft moonlight. |
| Figure 6 | A painting of brooklyn new york 1940 storefronts, by John Kay, highly textured, rich colour and detail, ballard, deep colour's, style of raymond swanland, trio, oill painting, h 768, well worn, displayed, detailed 4k oil painting, glenn barr, textured oil on canvas, looking cute. |
| Figure 7 | A gardener tending to a colorful, blooming garden, oil painting by Van Gogh. |
| | Portrait of a luxurious model in a plush coat. |
| Figure 8 | Furina from Genshin Impact sitting gracefully at a lavish dining table inside a grand palace hall. The setting is elegant and opulent, with towering arched windows letting in soft afternoon sunlight. The table is set for a classic English afternoon tea — delicate porcelain teacups, a silver teapot, and a multi-tiered tray filled with colorful macarons, scones with clotted cream and jam, finger sandwiches, and petits fours. Furina wears an ornate, Victorian-inspired outfit with lace gloves, looking both noble and contemplative. Crystal chandeliers hang overhead, casting a warm glow, and the palace decor is rich with gold accents and deep velvet drapes. |
| | Wukong sits cross-legged in deep meditation atop the blazing peaks of Flame Mountain. Surrounded by rivers of molten lava and pillars of rising smoke, his golden fur glows softly in the fiery light. His staff rests beside him, half-buried in scorched stone. Despite the searing heat and roaring flames around him, his expression is calm and focused, as if in perfect harmony with the chaos. The sky above is a storm of red and orange, casting a dramatic backdrop to his solitary training. |
| | A close-up portrait of a young woman with flawless skin, vibrant red lipstick, and wavy brown hair, wearing a vintage floral dress and standing in front of a blooming garden. |
| | Character of lion in style of saiyan, mafia, gangsta, citylights background, Hyper detailed, hyper realistic, unreal engine ue5, cgi 3d, cinematic shot, 8k. |
| | A jellyfish dances in the sea, against a backdrop of coral and seaweed. |
| | A product image of an iPhone standing upright in a forest. The iPhone's screen shows a view of the forest with a dramatic lightning strike captured in the background. The surrounding forest is lush and dense with trees, and the sky above is stormy and dark, with the lightning illuminating the scene. The background is clear and detailed, highlighting the contrast between the natural elements and the advanced technology of the phone. This image emphasizes the iPhone's high-quality display and camera capabilities, ideal for use in a tech blog or product design portfolio. |
| | A futuristic astronaut exploring an alien planet, with a detailed spacesuit and a landscape of strange plants and rock formations. |
| | A cat next to a stack of pancakes in a living room, best quality, extremely detailed, 8k. |
| | A frozen tundra with glowing ice spires and northern lights. |
| | Steampunk makeup, in the style of vray tracing, colorful impasto, dark cyan and amber makeup. Rich colourful plumes. Victorian style. |
| | Cinematic photo of delicious chocolate icecream. |
| Figure D.1 | A hamster piloting a tiny hot air balloon. |
| | A Samoyed wearing a sunglasses, sticking out its tongue, dslr image, 8k. |
| Figure E.2 | A potato astronaut in a spacesuit is watering a rose plant on the moon. It looks happy, Cyberpunk style. |
| Figure G.3 | A tiger is playing football. |
| Figure H.4 | Floating islands connected by waterfalls in a sunset sky. |
| | Summer landscape, vivid colors, a work of art, grotesque, Mysterious. |
| | A swirling night sky filled with bright stars and a small village below, inspired by Van Gogh's Starry Night. |
| | A frozen tundra with glowing ice spires and northern lights. |
| | A dreamlike landscape emerges from a first-person viewpoint, immersing the observer in an alluring world where waterlilies of soft lavender and violet hues gracefully drift on the surface of an opalescent pond. Towering lotus blossoms stretch towards an indigo sky embellished with celestial bodies that gleam like stars, invoking both tranquility and awe. A regal swan presides over this fantastical garden, its iridescent feathers creating captivating ripples across the water that seem to distort time itself, crafting a harmonious melody of dreams and nature, encapsulating the spirit of beauty and whimsy in one stunning tableau. |
| | A serene lakeside during autumn, with trees displaying a palette of fiery colors. |

Table H.3: The image generation prompts for each figure are listed sequentially, following the order from left to right and top to bottom. (Table 3/3)

| Figure | Text Prompt |
|---|---|
| Figure H.5 | A panoramic view of a city skyline at twilight, with skyscrapers and city lights. |
| | A long-exposure photograph of a vibrant city street at night, with light trails from moving cars. |
| | A deep forest clearing with a mirrored pond reflecting a galaxy-filled night sky. |
| | A desert oasis with palm trees and a shimmering pond. |
| | Primitive forest, towering trees, sunlight falling, vivid colors. |
| | A well-lit photograph of a modern, minimalist living room with large windows overlooking a city. |
| Figure H.6 | A classic still life composition featuring a teapot, teacup, flowers, and other decorative objects. |
| | Burning pile of money, epic composition, digital painting, emotionally profound, thought-provoking, intense and brooding tones, high quality, masterpiece. |
| | Create a hyper-realistic scene of a grand, classical hallway inside an opulent palace. The hallway is lined with towering columns and adorned with ornate, gilded paintings on the walls. Massive, powerful ocean waves surge through the corridor, crashing against the columns and splashing onto the walls. |
| | A jungle temple overgrown with vines and ancient carvings. |
| | A photograph of an abandoned building, showing decay and the interplay of light and shadow. |
| | A painting of brooklyn new york 1940 storefronts, by John Kay, highly textured, rich colour and detail, ballard, deep colour's, style of raymond swanland, trio, oill painting, h 768, well worn, displayed, detailed 4k oil painting, glenn barr, textured oil on canvas, looking cute. |
| Figure H.7 | A photorealistic, cinematic still from a historical drama depicting a battle-worn medieval knight amidst the ruins of a burning village at night. The scene is illuminated by flickering flames and the faint glow of embers, casting long, dancing shadows across the landscape. The knight's plate armor is heavily scarred and scratched, reflecting the firelight in fragmented patterns. |
| | A close-up portrait of a young woman with flawless skin, vibrant red lipstick, and wavy brown hair, wearing a vintage floral dress and standing in front of a blooming garden. |
| | Super detailed, selfie, college co-ed with pink hair in a pixie cut posed in front of her computer. Close up of face. brilliant blue eyes, full lips, pale skin and freckles. |
| | Portrait of man in a suit sitting in a gorgeous classical office. |
| | Cyberpunk hero with neon tattoos and futuristic armor. |
| | A charming, tech-savvy [girl with short, silver pixie-cut] hair and vibrant [blue] eyes, wearing a casual yet futuristic outfit. She's focused on a holographic interface while working in a sleek, high-tech workshop. |
| Figure H.8 | A hamster piloting a tiny hot air balloon. |
| | Digital art of a beautiful tiger under an apple tree, cartoon style, matte painting, magic realism, bright colors, hyper quality, high detail, high resolution. |
| | A cinematic photo of a cat with flower. |
| | A Samoyed wearing a sunglasses, sticking out its tongue, dslr image, 8k. |
| | A cute corgi cooking in the kitchen. |
| | A sea turtle swimming through a coral reef. |
| Figure H.9 | Astronaut in a jungle, cold color palette, muted colors, detailed, 8k. |
| | A woman in a pink dress walking down a street, cyberpunk art, inspired by Victor Mosquera, conceptual art, style of raymond swanland, yume nikki, restrained, robot girl, ghost in the shell. |
| | A quirky miniature scene, potato chip soldiers parachuting onto a ceramic bowl filled with ridged potato chips, tiny plastic figurines suspended by yellow mushroom cloud-like parachutes, surreal food photography, soft lighting. |
| | A DSLR shot of fresh strawberries in a ceramic bowl, with tiny water droplets on the fruit, highly detailed, sharp focus, photo-realistic, 8K. |
| | By Tang Yau Hoong, ultra hd, realistic, vivid colors, highly detailed, UHD drawing, pen and ink, perfect composition, beautiful detailed intricate insanely detailed octane render trending on artstation, 8k artistic photography, photorealistic concept art, soft natural volumetric cinematic perfect light, ultra hd, realistic, vivid colors, highly detailed, UHD drawing, pen and ink, perfect composition, beautiful detailed intricate insanely detailed octane render trending on artstation, 8k artistic photography, photorealistic concept art, soft natural volumetric cinematic perfect light. |
| | Gentleman cyborg robot with a fish bowl head, with a small goldfish. |
| Figure H.10 | Brown wooden house in the middle of snow covered trees. |
| | A sleek, high-performance supercar cruises through the bustling city streets. |
| Figure H.11 | A magnificent Gothic castle perched on a cliff, with waves crashing against the cliff walls, sunset over sea. |
| | A bird is singing on the branch with tender leaf, beside it is its nest, and the morning sunlight is shining. |
| | Close-up Advertisement, a floating pool table with a chilled bottle of Coca Cola, around with lemons and ice-cubes, all set against the bright sun and clear water. |
| | Photo of a rhino dressed suit and tie sitting at a table in a bar with a bar stools, award winning photography, Elke vogelsang. |
| | A beautiful woman, slightly bend and lower head, perfect face, pale red lips, Ultraviolet, Charlie Bowater style, Paper, The composition mode is waist shot style, Hopeful, Octane render, 4k HD, wearing a modest high-neck dress that elegantly covers the chest area, with intricate patterns reminiscent of traditional art. |
| | A heartwarming close-up scene featuring a playful kitten and a curious hedgehog sharing a cozy blanket. |
| | Einstein, a bronze statue, with a fresh red apple besides it, by Bruno Catalano. |
| | Astronaut on Mars During sunset. |

