# OpenReview forum: "HiFlow: Training-free High-Resolution Image Generation with Flow-Aligned Guidance"
_NeurIPS.cc/2025/Conference — NeurIPS 2025 poster_

### Official Review · Reviewer_iWGG · 2025-06-30

**Clarity:** 3
**Significance:** 3
**Originality:** 3
**Rating:** 5
**Confidence:** 4

**Summary:**

This paper addresses adapting pre-trained image diffusion/flow models for high-resolution image generation without further fine-tuning. The paper proposes to leverage a low-resolution reference image to guide the sampling of high-resolution via alignment at multiple granularities. While prior work anchors the guidance to a single upscaled prediction of the clean image, this work introduces timestep-dependent anchors to close the distribution gap between the guidance signal and the current noisy high-res input. The alignments altogether preserves the low-frequency structures of the low-res image while generating the high-frequency details with fidelity.

**Questions:**

1. How sensitive is the method to the weight hyperparameters used in all three alignments?
2. How robust is the method to low-resolution reference? Would artifacts and lack of details in the reference potentially propagate or constrain the generation of high-res image?

**Ethical Concerns:**

["NO or VERY MINOR ethics concerns only"]

**Final Justification:**

The authors have provided extensive results and discussions addressing reviewer concerns. The paper introduces a novel method for adapting pre-trained image diffusion models to generate at higher resolutions than training time. Their proposed approach outperforms all baselines and demonstrates significant qualitative improvement. I maintain my rating of recommending acceptance.

**Limitations:**

Yes, limitations and broader impacts are addressed in the appendix.

**Paper Formatting Concerns:**

No major formatting issues.

**Quality:**

3

**Strengths And Weaknesses:**

**Strengths**
1. The figures illustrate qualitative effects of each component of the method well and ablation is informative of the contribution of each alignment.
2. Baseline comparisons are comprehensive, covering both training-free and trained approaches as well as super-resolution models.
3. Qualitative results exhibit sharp high-resolution images capturing intricate patterns and details.
4. The effectiveness of the method when applied in conjunction with LoRA and ControlNet, quantized model setting, and across different architectures(UNet/DiT) is well demonstrated.


**Weaknesses**

1. The figure outlining the pipeline is not straightforward to follow and makes it hard to grasp the essence of the method. Arrows eventually pointing to acceleration can be misleading. Illustration can be clarified further.
2. Although $\alpha$ and $\beta$ parameters follow a gradually weakening schedule according to t/$\tau$, this choice of function and its impact on parameters and image outputs of different resolutions are not extensively discussed. Specific choices for these schedules could potentially limit the effectiveness of the guidance.
3. More discussion about achieving fidelity with respect to the reference can be added. For instance in Fig. 5, although the palm tree and owl zoom-ins present more visual details, they are not consistent with the reference.

---

> ### Author Rebuttal · Authors · 2025-07-30
>
> We are very encouraged to see that you found our work effective and conducted comprehensive experiments. We sincerely thank you for your valuable suggestions, which certainly help improve our work. We have accordingly refined our paper as follows:
>
> ---
>
> ### **Q1: The figure outlining the pipeline is not straightforward to follow and makes it hard to grasp the essence of the method. Arrows eventually pointing to acceleration can be misleading. Illustration can be clarified further.**
>
> **A1:**  Thanks for your valuable suggestions, and we sincerely apologize for any misunderstanding that may have been caused. The acceleration depicted in the pipeline figure is merely for visualization purposes, and the flow arrows should not eventually point to acceleration, as this could be misleading. We will comprehensively restructure the pipeline diagram in future versions for better presentation.
>
> ### **Q2: The choice of guidance weight decay function and its impact on parameters and image outputs of different resolutions are not extensively discussed. How sensitive is the method to the weight hyperparameters used in all three alignments?**
>
> **A2:** Thanks for your constructive advice.  We have supplemented comprehensive sensitivity analysis experiments regarding guidance weight strategies (direction guidance weight $\alpha$ and acceleration guidance weight $\beta$) and other hyperparameters (noise adding ratio $\tau$ and normalized cutoff frequency $D$) at 2K and 4K resolution, as shown in the following tables.
>
> For 2K generation, linear decay and cosine decay have similar performance, while for 4K generation, linear decay slightly outperforms cosine decay. We speculate that this is because the noise adding ratio is lower in the 4K generation, and the guidance strength of cosine decay may be insufficient at this lower noise level. Overall, both linear and cosine decay strategies enable gradually weakening control, significantly outperforming the constant guidance weight strategy without any decay. In our experiments, we used linear decay for both 2K and 4K.
>
> Furthermore, we have also discussed the choices for noise adding ratio $\tau$ and normalized cutoff frequency $D$. For 2K generation, a relatively wide range of $\tau$ (e.g., $\tau\in[0.5,0.7]$) yields similar results. However, larger $\tau$ entails higher latency. Considering both performance and time cost, $\tau=0.6$ is chosen. For 4K generation, given the unstable synthesis capability of the backbone model, a smaller noise adding ratio ($\tau\in[0.2, 0.3]$) is more suitable. For both 2K and 4K generation, a moderate normalized cutoff frequency (e.g., $D\in[0.4,0.5]$) is recommended, as such a frequency effectively captures the image layout without carrying low-resolution detail information. In our experiments, we used $\tau=0.6$ for 2K, $\tau=0.3$ for 4K, and $D=0.4$ for both 2K and 4K.
>
> **Table: Quantitative evaluation on guidance weight strategies $\alpha$ and $\beta$ (2K resolution).**
>
> | **Metrics-2K**       | **Constant Value (0.5)** | **Linear Decay ($t/τ$)** | **Cosine Decay ($0.5 × (1 - cos(πt/τ))$)** |
> |:--------------------:|:------------------------:|:------------------------:|:------------------------------------------:|
> | **FID**              | 58.65                    | **55.39**                | 55.77                                      |
> | **FID_patch**        | 51.05                    | **47.70**                | 47.79                                      |
> | **IS**               | 26.82                    | 28.67                    | **28.75**                                  |
> | **IS_patch**         | 12.77                    | 13.86                    | **13.89**                                  |
> | **CLIP Score**       | 34.75                    | 35.32                    | **35.39**                                  |
>
> **Table: Quantitative evaluation on guidance weight strategies $\alpha$ and $\beta$ (4K resolution).**
>
> | **Metrics-4K**       | **Constant Value (0.5)** | **Linear Decay ($t/τ$)** | **Cosine Decay ($0.5 × (1 - cos(πt/τ))$)** |
> |:--------------------:|:------------------------:|:------------------------:|:------------------------------------------:|
> | **FID**              | 55.59                    | **52.55**                | 53.03                                      |
> | **FID_patch**        | 50.21                    | **45.01**                | 47.09                                      |
> | **IS**               | 23.12                    | 24.62                    | **24.88**                                  |
> | **IS_patch**         | 8.63                     | **9.73**                 | 9.54                                       |
> | **CLIP Score**       | 34.92                    | **35.40**                | 35.27                                      |
>
> **Table: Quantitative evaluation on hyperparameters $\tau$ and $D$ (2K resolution).**
> | **Metrics-2K**        |   $\tau=0.5$   |   $\tau=0.6$   |   $\tau=0.7$   |   $D=0.3$   |   $D=0.4$    |   $D=0.5$    |
> |:----------------------:|:--------------:|:-----------:|:-----------:|:--------------:|:-----------:|:-----------:|
> | **FID**               | 56.12          | 55.39       | **55.28**   | 55.91          | **55.39**   | 55.45       |
> | **FID_patch**         | 48.49          | 47.70       | **47.45**   | 48.42          | **47.70**   | 48.09       |
> | **IS**                | 28.62          | **28.67**   | 28.55       | 28.44          | 28.67       | **28.72**   |
> | **IS_patch**          | 13.80          | **13.86**   | 13.74       | 13.79          | 13.86       | **13.89**   |
> | **CLIP Score**        | 35.24          | 35.32       | **35.37**   | 35.19          | **35.32**   | 35.28       |
>
> **Table: Quantitative evaluation on hyperparameters $\tau$ and $D$ (4K resolution).**
> | **Metrics-4K**        | $\tau=0.2$   | $\tau=0.3$      | $\tau=0.4$      | $D=0.3$    | $D=0.4$     | $D=0.5$     |
> |:---------------------:|:------------:|:-----------:|:-----------:|:------------:|:-----------:|:-----------:|
> | **FID**               | **52.36**    | 52.55       | 54.39       | 52.83        | 52.55       | **52.47**   |
> | **FID_patch**         | **44.93**    | 45.01       | 48.40       | 45.82        | **45.01**   | 46.43       |
> | **IS**                | 24.56        | **24.62**   | 23.78       | 23.50        | **24.62**   | 24.41       |
> | **IS_patch**          | 9.46         | **9.73**    | 8.85        | 9.34         | **9.73**    | 9.69        |
> | **CLIP Score**        | 35.33        | **35.40**   | 35.26       | 35.32        | 35.40       | **35.42**   |
>
> ### **Q3: More discussion about achieving fidelity with respect to the reference can be added. For instance in Fig. 5, although the palm tree and owl zoom-ins present more visual details, they are not consistent with the reference.**
>
> **A3:** Thank you for your valuable advice. Previous high-resolution generation works (e.g., DemoFusion and DiffuseHigh) typically enforced alignment of each step on their sampling trajectory with the same low-resolution endpoint. Despite higher similarity with the reference, they usually suffer from a limited amount of new details and risk generating artifacts and low-fidelity details. In contrast, HiFlow leverages the step-wise information of the low-resolution flow for guided generation, without mandating consistency with the low-resolution result. **Instead of copying the low-resolution information, HiFlow effectively extends the model's synthesis capability learned at the training resolution to the high-resolution space through the proposed flow-aligned guidance**, achieving high-resolution image generation with high fidelity and rich details.
>
> ### **Q4: How robust is the method to low-resolution reference? Would artifacts and lack of details in the reference potentially propagate or constrain the generation of high-res image?**
>
> **A4:** Thanks for your constructive comment. **HiFlow is relatively robust to local artifacts in the low-resolution flow but may fail when facing severe structural anomalies.**
>
> It is observed that local artifacts in low-resolution images can be corrected in our high-resolution results.
> For instance, please refer to the "street" case in the lower right corner of Fig.F.5 in the appendix. In its 1K result (will be added in future versions), the items in the store window are blurry artifacts. However, in the 4K result generated by HiFlow, the artifacts are replaced with reasonable product details.
> This is because HiFlow leverages the low-resolution trajectory for flow-aligned guidance, which is a guided generation process rather than directly copying low-resolution information, allowing for the generation of new, reasonable details to replace artifacts. Additional visual results on artifact correction will be added in future versions.
>
> However, if there exist significant structural anomalies (such as duplicate limbs) in the low-resolution results, the high-resolution results may inherit these errors. This stems from the limited performance of the image generation backbone.
>
> **Fortunately, given HiFlow's training-free and model-agnostic nature, such flaws can be effectively mitigated** by integrating HiFlow into a stronger backbone model, using ControlNet to introduce external control conditions for controllable generation (see Fig.8 (b) in the main paper), or applying inference-time scaling techniques to search for a better low-resolution trajectory for the subsequent high-resolution generation.
>
> Moreover, **the lack of details in the reference typically does not affect the high-resolution generation**. Consider the synthesis process from 1K to 4K, where 15 times more new content is generated, far exceeding the original, allowing for the addition of more details.  However, the style specified by the prompt can have some impact on the amount of detail.  For example, the comic/anime style will have slightly fewer details compared to those in the realistic style.

---

> ### Comment · Reviewer_iWGG · 2025-08-05
>
> Thank you for sharing additional results verifying the effects of hyperparameters and weight decay functions across resolutions. The provided tables and discussion help clarify the design choices. Given that the method offers training-free adaption to high-resolution generations through bootstrapping on its low-resolution sample while improving details and being robust to artifacts, I maintain my original rating to recommend acceptance.

---

### Official Review · Reviewer_htDR · 2025-06-30

**Clarity:** 3
**Significance:** 3
**Originality:** 3
**Rating:** 5
**Confidence:** 2

**Summary:**

This paper introduces HiFlow, a training-free and model-agnostic framework for high-resolution image generation using pre-trained diffusion/flow models. HiFlow addresses the challenge of generating high-quality, high-resolution images by creating a virtual reference flow in the high-resolution space. It guides the generation process through three key strategies: initialization alignment (for low-frequency consistency), direction alignment (for structure preservation), and acceleration alignment (for detail fidelity). By leveraging these flow-aligned techniques, HiFlow significantly improves the quality of high-resolution image synthesis and works effectively across various personalized T2I models. Experimental results show that HiFlow outperforms existing state-of-the-art methods in generating high-resolution images.

**Questions:**

Please see Weaknesses.

The acceleration visualizations presented in Figure 3 (c) and Figure 4 are somewhat unclear. Could the authors provide more explanation or interpretation regarding what these visualizations represent and how they support the claims about detail fidelity?

Additionally, can the three techniques proposed by HiFlow be well applied to high-resolution video generation tasks? It would be helpful if the author could briefly discuss this.

**Ethical Concerns:**

["NO or VERY MINOR ethics concerns only"]

**Final Justification:**

The authors have addressed all of my concerns in the rebuttal, including acceleration visualizations, evaluation datasets, and the extension to high-resolution video generation tasks. I sincerely appreciate the thorough and comprehensive response. Based on this, I recommend accepting the paper.

**Limitations:**

yes

**Quality:**

3

**Strengths And Weaknesses:**

1. The proposed HiFlow is simple and effective, and it is very easy to understand.

2. Low-frequency consistency, structure preservation, and detail fidelity are crucial for high-resolution images. The author has designed corresponding modules for each of these aspects, making the overall framework design very reasonable.

3. HiFlow achieves high-quality visual results and outperforms most previous methods in quantitative evaluations.

4. The acceleration visualization in Figure 3 (c) and also in Figure 4 is a bit confusing. It's not very clear what the author intends to convey with these illustrations.

5. The evaluation dataset has not been specified or discussed. Additionally, there are relatively few quantitative evaluation experiments. It appears that the author has conducted ablation and comparison assessments on only a single evaluation dataset, which may not provide a comprehensive view of HiFlow's effectiveness.

---

> ### Author Rebuttal · Authors · 2025-07-30
>
> We are highly encouraged to see that you found our method effective, reasonable, and easy to understand. We sincerely thank you for your valuable suggestions, which certainly help improve our work. We have accordingly refined our paper as follows:
>
> ---
>
> ### **Q1: The acceleration visualizations are unclear. Could the authors provide more explanation or interpretation regarding what these visualizations represent and how they support the claims about detail fidelity?**
>
> **A1:**  Thank you for your valuable advice, and we sincerely apologize for any misunderstanding that may have been caused. As we discussed in Section 3.3 in the main paper, the flow acceleration $a_t$ can eventually be expressed as:
>
> $$a_t = -\frac{1}{t_i}\frac{X_{0\leftarrow t_{i-1}}-X_{0\leftarrow t_i}}{t_{i-1}-t_i}=-\frac{1}{t_i}\frac{dX_{0\leftarrow t}}{dt},$$
>
> which indicates that the acceleration term $a_t$ essentially depicts the first order derivative of the estimated clean sample $X_{0\leftarrow t}$ with respect to timestep $t$, multiplied by a time-dependent factor $-1/t$. Note that the estimated clean sample $X_{0\leftarrow t}$ stands for the noise-free component of current noisy latent $X_t$, i.e. the expected clean image at timestep $t$. Therefore, $a_t$ can serve as an indicator of the sequence of content synthesized at each timestep, capturing the difference between estimated clean samples. In other words, $a_t$ reflects what content the model is responsible for adding at different timestep $t$. **We visualize $a_t$ using the jet map from matplotlib, as shown in Fig.D.1 in the appendix, with the results clearly reflecting the model's synthesis content sequence**. Leveraging acceleration alignment, we explicitly guide the model on what content to synthesize at each timestep in high-resolution generation, thereby avoiding the generation of unrealistic contents like repetitive patterns and abnormal textures and facilitating higher detail fidelity.
>
> ### **Q2: The evaluation dataset has not been specified or discussed. Only a single evaluation dataset is used in quantitative experiments, which may not provide a comprehensive view of HiFlow's effectiveness.**
>
> **A2:** Thanks for your valuable suggestions, and we sincerely apologize for any unclear descriptions regarding our evaluation dataset.
> **The dataset we used contains 1K high-quality captions, primarily collected from previous works on high-resolution image generation (e.g., UltraPixel [1], I-Max [2]) and open-source datasets (e.g., LAION-400M [3], Aesthetic-4K [4]). Then, GPT-4o was employed to filter and polish them to improve quality**. Eventually, our dataset covers a diverse range of generation scenarios and provides precise descriptions of image details. A brief comparison of the caption quality between our self-collected dataset and the open-sourced LAION-400M dataset is shown below:
>
> Captions sampled from LAION-400M:
>
> *"Khoni Makan Urdu Novel by A Hameed"*
>
> *"geisha+hairstyle+portrait+2"*
>
> Captions sampled from our evaluation dataset:
>
> *"A blood-red Ferrari SF90 parked on a rain-soaked city street at night, reflecting neon lights from nearby buildings. The wet pavement glistens, and the car’s smooth curves are highlighted by the ambient glow of the urban environment"*
>
> *"Digital art of a beautiful tiger under an apple tree, cartoon style, matte painting, magic realism, bright colors, hyper quality, high detail, high resolution."*
>
> For a more comprehensive assessment, we have **randomly sampled 1K captions from the LAION-5B [5] dataset to construct a new benchmark** for quantitative evaluation and ablation study. The results are shown in the following tables. Please note that, due to the suboptimal quality of captions in the LAION-5B dataset, the CLIP Score for all tested methods has declined compared to our self-collected benchmark.
>
> **Table: Quantitative Evaluation on LAION-5B Benchmark.**
>
> | **Metrics-4K**     | **FID**       | **FID_patch**     | **IS**         | **IS_patch**     | **CLIP Score**   |
> |:------------------:|:-------------:|:-----------------:|:--------------:|:----------------:|:----------------:|
> | DemoFusion     | 59.01   | 49.22     | 20.02   | 8.38     | 29.92       |
> | DiffuseHigh    | 60.24   | 50.73     | 20.41   | 8.06     | 28.36       |
> | I-Max          | *55.96* | 53.25     | 21.74   | 7.58     | 30.12       |
> | Flux + BSRGAN  | 60.87   | 55.71     | 19.67   | 8.74     | **30.77**   |
> | UltraPixel     | 58.11   | *48.02*   | **24.21** | *9.20*   | 29.53       |
> | Diffusion-4k   | 76.48   | 72.27     | 15.41   | 6.57     | 25.85       |
> | HiFlow (Ours)  | **54.78** | **45.80** | *24.07* | **9.52** | *30.65*     |
>
> **Table: Ablation Study on LAION-5B Benchmark.**
>
> | **Metrics-4K**         | **FID**     | **FID_patch** | **IS**        | **IS_patch** | **CLIP Score** |
> |:----------------------:|:-----------:|:-------------:|:-------------:|:------------:|:--------------:|
> | $-A_a, -A_d, -A_i$     | 227.98    | 200.03    | 9.12      | 3.04     | 11.23       |
> | $-A_a, -A_d$           | 57.05     | 49.34     | 18.87     | 6.24     | 28.60       |
> | $-A_a$                 | *56.41*   | *48.90*   | *22.33*   | *8.81*   | *30.09*     |
> | HiFlow (Ours)          | **54.78** | **45.80** | **24.07** | **9.52** | **30.65**   |
>
> ### **Q3: Can the three techniques proposed by HiFlow be well applied to high-resolution video generation tasks?**
>
> **A3:** Thank you for your constructive suggestions. We have validated the potential of HiFlow on the state-of-the-art T2V model WAN 2.1 [6], in which HiFlow exhibits remarkable effectiveness in training-free high-resolution video generation with flow-aligned guidance. Specifically, we integrate HiFlow with WAN 2.1-1.3B to generate videos at a resolution of $2080\times1200$ and set the number of frames to $33$ to save GPU memory. As measured by VBench [7] and CLIP [8] Score, HiFlow significantly improves WAN 2.1's generation capability at high resolution, especially in terms of visual quality and text-image similarity. Quantitative results are listed below. Please note that WAN 2.1 usually experiences generation collapse when directly inferring high-resolution videos, resulting in semantically void static videos, which is why its background consistency is slightly higher than HiFlow.
>
> | **Evaluation Metric**         | **Subject Consistency** | **Background Consistency** | **Motion Smoothness** | **Dynamic Degree** | **Aesthetic Quality** | **Imaging Quality** | **CLIP Score** |
> |:-----------------------------:|:--------------------------:|:-----------------------------:|:------------------------:|:---------------------:|:------------------------:|:----------------------:|:----------------:|
> | WAN 2.1 (Direct Inference 4K) | 0.9348                     | **0.9578**                    | 0.9823                   | 0.6489                | 0.4728                   | 0.4109                 | 14.89            |
> | WAN 2.1 + HiFlow              | **0.9394**                 | 0.9521                        | **0.9827**               | **0.7900**            | **0.5891**               | **0.6153**             | **32.75**        |
>
> ### **Reference**
>
> [1] Ren J, Li W, Chen H, et al. Ultrapixel: Advancing ultra high-resolution image synthesis to new peaks[J]. Advances in Neural Information Processing Systems, 2024, 37: 111131-111171.
>
> [2] Du R, Liu D, Zhuo L, et al. I-max: Maximize the resolution potential of pre-trained rectified flow transformers with projected flow[J]. 2024.
>
> [3] Schuhmann C, Vencu R, Beaumont R, et al. Laion-400m: Open dataset of clip-filtered 400 million image-text pairs[J]. arXiv preprint arXiv:2111.02114, 2021.
>
> [4] Zhang J, Huang Q, Liu J, et al. Diffusion-4k: Ultra-high-resolution image synthesis with latent diffusion models[C]//Proceedings of the Computer Vision and Pattern Recognition Conference. 2025: 23464-23473.
>
> [5] Schuhmann C, Beaumont R, Vencu R, et al. Laion-5b: An open large-scale dataset for training next generation image-text models[J]. Advances in neural information processing systems, 2022, 35: 25278-25294.
>
> [6] Wan T, Wang A, Ai B, et al. Wan: Open and advanced large-scale video generative models[J]. arXiv preprint arXiv:2503.20314, 2025.
>
> [7] Huang Z, He Y, Yu J, et al. Vbench: Comprehensive benchmark suite for video generative models[C]//Proceedings of the IEEE/CVF Conference on Computer Vision and Pattern Recognition. 2024: 21807-21818.
>
> [8] Radford A, Kim J W, Hallacy C, et al. Learning transferable visual models from natural language supervision[C]//International conference on machine learning. PmLR, 2021: 8748-8763.

---

> > ### Comment · Reviewer_htDR · 2025-08-02
> >
> > Thank you for the detailed and substantial response. My concerns have been fully addressed, and I am inclined to recommend acceptance based on the current state of the submission.

---

> > > ### Author Response · Authors · 2025-08-02
> > >
> > > Dear Reviewer htDR,
> > >
> > > Thanks for your positive feedback on our work!
> > >
> > > I'm glad to hear that our rebuttal has addressed your concerns. We sincerely appreciate your time and effort in providing such meticulous reviews and insightful comments.
> > >
> > > Best regards,
> > >
> > > Authors

---

### Official Review · Reviewer_7Jea · 2025-07-02

**Clarity:** 3
**Significance:** 3
**Originality:** 3
**Rating:** 5
**Confidence:** 3

**Summary:**

The paper presents HiFlow, a novel training-free framework for high-resolution image synthesis using pre-trained flow models. The approach is well-motivated and addresses a significant challenge in text-to-image generation. The methodology is sound, and the experimental results demonstrate clear improvements over existing methods.

**Questions:**

Please refer to the paper weakness

**Ethical Concerns:**

["NO or VERY MINOR ethics concerns only"]

**Limitations:**

The failure case in Figure E.2 (duplicated tail artifact) suggests HiFlow inherits structural flaws from low-resolution inputs. The authors should discuss mitigation strategies (e.g., post-processing or iterative refinement).

**Quality:**

3

**Strengths And Weaknesses:**

**Paper Strengths**

Novelty and Contribution:
1)	The introduction of virtual reference flow and flow-aligned guidance (initialization, direction, and acceleration alignment) is innovative. The idea of leveraging intermediate states of low-resolution sampling trajectories for high-resolution synthesis is compelling.
2)	The framework is model-agnostic and compatible with architectures like U-Net and DiT, enhancing its applicability.

Experimental Results: Quantitative metrics (FID, IS, CLIP score) and qualitative comparisons (Figures 5–6) convincingly show HiFlow’s superiority over baselines, including training-based methods like UltraPixel and Diffusion-4k.

**Paper Weaknesses**

Acceleration Alignment: The physical interpretation of acceleration (Eq. 10) is insightful but could be better motivated. How does aligning acceleration empirically improve detail fidelity compared to direction alignment alone?

Hyperparameters: The choice of noise-adding ratio τ and cutoff frequency D seems heuristic. A sensitivity analysis would justify these choices.

---

> ### Author Rebuttal · Authors · 2025-07-30
>
> We are very encouraged to see that you found our work innovative, sound and addressing a significant challenge in text-to-image generation. We sincerely thank you for your valuable suggestions, which certainly help improve our work. We have accordingly refined our paper as follows:
>
> ---
>
> ### **Q1: Motivation of acceleration alignment, how does aligning acceleration empirically improve detail fidelity compared to direction alignment alone?**
>
> **A1:**  Thanks for your valuable suggestions. Equipped with initialization alignment and direction alignment, we are able to generate high-resolution images with rich details and reasonable structures. However, it is observed that the synthesized details are often low-fidelity and unrealistic (see Fig.3 (d) in the main paper). To address these issues, we analyze the flow velocity $v_t$ at each timestep. It indicates the denoising direction of each step in the sampling trajectory, thereby determining the position of the next state. The emergence of low-fidelity details suggests that the content synthesized by the high-resolution flow deviates from the normal content synthesis distribution. This stems from the flow model's unstable performance in high-resolution generation.
>
> To this end, **we consider utilizing the rate of change of velocity (which represents acceleration in physics) to control the amount of change in velocity at each timestep within a reasonable range, thereby avoiding suboptimal denoising directions caused by unreasonable velocity predictions during high-resolution generation**. Such a guidance signal is provided by the acceleration of the reference flow at each timestep, as it reflects the detail synthesis distribution learned by the model at its training resolution.
>
> Through further derivation of the acceleration (detailed in the main paper), we uncover that the acceleration term $a_t$ can be expressed as:
> $$a_t = -\frac{1}{t_i}\frac{X_{0\leftarrow t_{i-1}}-X_{0\leftarrow t_i}}{t_{i-1}-t_i}=-\frac{1}{t_i}\frac{dX_{0\leftarrow t}}{dt},$$
> which indicates that the acceleration term $a_t$ essentially depicts the first order derivative of the estimated clean sample $X_{0\leftarrow t}$ with respect to timestep $t$, multiplied by a time-dependent factor $-1/t$. Therefore, $a_t$ can serve as an indicator of the sequence of content synthesized at each timestep, capturing the difference between estimated clean samples. In other words, $a_t$ reflects what content the model is responsible for adding at different timesteps.
>
> **Motivated by the above analysis and derivation, we adopt acceleration alignment  in high-resolution generation to guide the model on what content to synthesize at each timestep, thereby avoiding the generation of unrealistic contents and facilitating higher detail fidelity**. In contrast, using only direction alignment does not achieve the same effect, as shown in the visual ablation results in Fig.7 in the main paper.
>
> ### **Q2: Sensitivity Analysis of Hyperparameters.**
>
> **A2:** Thanks for your constructive suggestions. We have supplemented comprehensive sensitivity analysis experiments regarding hyperparameters (noise adding ratio $\tau$ and normalized cutoff frequency $D$) at 2K and 4K resolution, as shown in the following tables.
>
> For 2K generation, a relatively wide range of $\tau$ (e.g., $\tau\in[0.5,0.7]$) yields similar results. However, larger $\tau$ entails higher latency. Considering both performance and time cost, $\tau=0.6$ is chosen. For 4K generation, given the unstable synthesis capability of the backbone model, a smaller noise adding ratio ($\tau\in[0.2, 0.3]$) is more suitable. For both 2K and 4K generation, a moderate normalized cutoff frequency (e.g., $D\in[0.4,0.5]$) is recommended, as such a frequency effectively captures the image layout without carrying low-resolution detail information. In our experiments, we adopted $\tau=0.6$ for 2K generation, $\tau=0.3$ for 4K generation, and $D=0.4$ for both 2K and 4K generation.
>
> **Table: Quantitative evaluation on hyperparameters (2K resolution).**
> | **Metrics-2K**        |   $\tau=0.5$   |   $\tau=0.6$   |   $\tau=0.7$   |   $D=0.3$   |   $D=0.4$    |   $D=0.5$    |
> |:----------------------:|:--------------:|:-----------:|:-----------:|:--------------:|:-----------:|:-----------:|
> | **FID**               | 56.12          | 55.39       | **55.28**   | 55.91          | **55.39**   | 55.45       |
> | **FID_patch**         | 48.49          | 47.70       | **47.45**   | 48.42          | **47.70**   | 48.09       |
> | **IS**                | 28.62          | **28.67**   | 28.55       | 28.44          | 28.67       | **28.72**   |
> | **IS_patch**          | 13.80          | **13.86**   | 13.74       | 13.79          | 13.86       | **13.89**   |
> | **CLIP Score**        | 35.24          | 35.32       | **35.37**   | 35.19          | **35.32**   | 35.28       |
>
> **Table: Quantitative evaluation on hyperparameters (4K resolution).**
> | **Metrics-4K**        | $\tau=0.2$   | $\tau=0.3$      | $\tau=0.4$      | $D=0.3$    | $D=0.4$     | $D=0.5$     |
> |:---------------------:|:------------:|:-----------:|:-----------:|:------------:|:-----------:|:-----------:|
> | **FID**               | **52.36**    | 52.55       | 54.39       | 52.83        | 52.55       | **52.47**   |
> | **FID_patch**         | **44.93**    | 45.01       | 48.40       | 45.82        | **45.01**   | 46.43       |
> | **IS**                | 24.56        | **24.62**   | 23.78       | 23.50        | **24.62**   | 24.41       |
> | **IS_patch**          | 9.46         | **9.73**    | 8.85        | 9.34         | **9.73**    | 9.69        |
> | **CLIP Score**        | 35.33        | **35.40**   | 35.26       | 35.32        | 35.40       | **35.42**   |
>
>
> ### **Q3: Mitigation strategies for structural flaws from low-resolution inputs.**
>
> **A3:** Thank you for your valuable advice. Here, we provide **three simple and effective methods** to reduce the potential structural error in the low-resolution flow, thereby fundamentally preventing error propagation:
>
> 1. Since HiFlow is training-free and model-agnostic, it can be seamlessly adapted to **a stronger image generation backbone**, thereby reducing the probability of structural anomalies in the low-resolution flow. For example, HiFlow exhibits far fewer structural anomalies on Flux compared to SDXL.
>
> 2. HiFlow can be combined with **controllable generation techniques** such as ControlNet [1] (see Fig.8 (b) in the main paper). Therefore, ControlNet can be utilized to introduce structural control signals to avoid potential structural anomalies.
>
> 3. **Inference time scaling methods** [2] can be adopted to improve the quality of low-resolution results by searching for a better initial noise (to obtain a better low-resolution flow) for the subsequent high-resolution generation. They typically include an auxiliary evaluation model (such as a MLLM or a specific verifier model like ImageReward [3]) to assess the output quality and iteratively search for the next noise in the neighborhood of already searched high-quality noises. Since the evaluation is conducted at low resolution (1K), this approach does not entail a significant time cost.
>
> ### **Reference**
>
> [1] Zhang L, Rao A, Agrawala M. Adding conditional control to text-to-image diffusion models[C]//Proceedings of the IEEE/CVF international conference on computer vision. 2023: 3836-3847.
>
> [2] Ma N, Tong S, Jia H, et al. Inference-time scaling for diffusion models beyond scaling denoising steps[J]. arXiv preprint arXiv:2501.09732, 2025.
>
> [3] Xu J, Liu X, Wu Y, et al. Imagereward: Learning and evaluating human preferences for text-to-image generation[J]. Advances in Neural Information Processing Systems, 2023, 36: 15903-15935.

---

> > ### Author Response · Authors · 2025-08-07
> >
> > Dear Reviewer 7Jea,
> >
> > We sincerely appreciate your time and effort in providing such meticulous reviews and insightful comments.
> >
> > Could you please kindly let us know if our rebuttal has addressed your concerns?
> >
> > If you have any further questions regarding our work, we would be delighted to address them.
> >
> > Best regards,
> >
> > Authors

---

> > ### Comment · Reviewer_7Jea · 2025-08-08
> >
> > Thank you for the detailed and substantial response. My concerns have been fully addressed. I will keep my positive rating.

---

> > > ### Author Response · Authors · 2025-08-08
> > >
> > > Dear Reviewer 7Jea,
> > >
> > > Thanks for your positive feedback on our work!
> > >
> > > I'm glad to hear that our rebuttal has addressed your concerns. We truly appreciate the time and effort you've dedicated to the review process.
> > >
> > > Best regards,
> > >
> > > Authors

---

### Official Review · Reviewer_b5PA · 2025-07-02

**Clarity:** 3
**Significance:** 3
**Originality:** 3
**Rating:** 5
**Confidence:** 3

**Summary:**

The paper addresses the problem of low quality in high-resolution image generation. The paper introduces HiFlow, which is a training-free and model-agnostic framework using pre-trained rectified flow models. In detail, HiFlow uses the low-resolution generation trajectory to construct a virtual reference flow. Then it is used to guide the high-resolution generation. There are three alignment mechanisms: initialization alignment (consistent low-frequency components), direction alignment (structural preservation), and acceleration alignment (high-fidelity detail). The method is evaluated on several benchmarks, and the experiments show that the proposed method achieves better performance compared to other baselines in terms of FID, IS, and CLIPScore.

**Questions:**

1. Can the authors elaborate on how robust the virtual reference flow is to artifacts in the low-resolution generation? Is there any method to reduce the risk of error propagation?
2. There are some acceleration sampling methods for diffusion or flow methods. Can they be used in the proposed method to accelerate the generation?
3. Is there any detection method to prevent errors in the alignments?

**Ethical Concerns:**

["NO or VERY MINOR ethics concerns only"]

**Final Justification:**

The authors have adequately addressed my concerns regarding artifacts in low-resolution generation, computational cost, and error detection. Therefore, I have decided to raise my score.

**Limitations:**

Yes

**Quality:**

3

**Strengths And Weaknesses:**

Strengths:
1. The motivation is reasonable. Now, the low-quality issue in high-resolution image generation is important. The paper aims to improve high-resolution generation without requiring retraining compared to previous methods.
2. The method is mode-agnostic and training-free. The method can be used in different architectures, such as DiT and U-net. It can also be applied to quantized models and personalization modules such as LoRA and ControlNet. It does not need any model-specific tuning.
3. The method is reasonable. The paper introduces a virtual reference flow and three flow-aligned guidance to improve the high-resolution generation. The method is interpretable and effective.
4. There are plenty of experiments in the paper. The authors conduct lots of experiments on different benchmarks and ablation studies to validate the proposed method.

Weaknesses:
1. The method heavily depends on low-resolution trajectory quality. The method points out that some previous methods are based on low-resolution image quality. However, the method assumes the low-resolution generation is structurally accurate. The artifact or mistakes in the low-resolution image have a bad influence on the high-resolution generation.
2. The method requires high memory and computational cost. The method needs 379s on A100 to generate 4k images, which is very computationally expensive. It limits the deployment.

---

> ### Author Rebuttal · Authors · 2025-07-30
>
> We are very encouraged to see that you found our method reasonable, interpretable and effective. We sincerely thank you for your valuable suggestions, which certainly help improve our work. We have accordingly refined our paper as follows:
>
> ---
>
> ### **Q1: HiFlow depends on low-resolution trajectory quality. The artifact or mistakes in the low-resolution image can have a bad influence on the high-resolution generation.**
> **A1:** Thanks for your constructive comment. **HiFlow is relatively robust to local artifacts in the low-resolution flow but may fail when facing severe structural anomalies.**
>
> It is observed that local artifacts in low-resolution images can be corrected in our high-resolution results.
> For instance, please refer to the "street" case in the lower right corner of Fig.F.5 in the appendix. In its 1K result (will be added in future versions), the items in the store window are blurry artifacts. However, in the 4K result generated by HiFlow, the artifacts are replaced with reasonable product details. This is because HiFlow leverages the low-resolution trajectory for flow-aligned guidance, which is a guided generation process rather than directly copying low-resolution information, allowing for the generation of new, reasonable details to replace artifacts. Additional visual results on artifact correction will be added in future versions.
>
> However, if there exist significant structural anomalies (such as duplicate limbs) in the low-resolution results, the high-resolution results may inherit these errors. This stems from the limited performance of the image generation backbone.
>
> **Fortunately, given HiFlow's training-free and model-agnostic nature, such flaws can be effectively mitigated** by integrating HiFlow into a stronger backbone model, using ControlNet [1] to introduce external control conditions for controllable generation (see Fig.8 (b) in the main paper), or applying inference-time scaling techniques [2] to search for a better low-resolution trajectory for the subsequent high-resolution generation.
>
> ### **Q2: How robust the virtual reference flow is to artifacts in the low-resolution generation? Is there any method to reduce the risk of error propagation?**
> **A2:** Thank you for your valuable suggestion. As we discussed in the answer to Question 1, **HiFlow (which obtains guidance from the virtual reference flow) is relatively robust to local artifacts but is not robust to severe structural anomalies**.
>
> Here, we provide **three simple and effective methods** to reduce the potential structural error in the low-resolution flow, thereby fundamentally preventing error propagation:
>
> 1. Since HiFlow is training-free and model-agnostic, it can be seamlessly adapted to **a stronger image generation backbone**, thereby reducing the probability of structural anomalies in the low-resolution flow. For example, HiFlow exhibits far fewer structural anomalies on Flux compared to SDXL.
>
> 2. HiFlow can be combined with **controllable generation techniques** such as ControlNet [1] (see Fig.8 (b) in the main paper). Therefore, ControlNet can be utilized to introduce structural control signals to avoid potential structural anomalies.
>
> 3. **Inference time scaling methods** [2] can be adopted to improve the quality of low-resolution results by searching for a better initial noise (to obtain a better low-resolution flow) for the subsequent high-resolution generation. They typically include an auxiliary evaluation model (such as a MLLM or a specific verifier model like ImageReward [3]) to assess the output quality and iteratively search for the next noise in the neighborhood of already searched high-quality noises. Since the evaluation is conducted at low resolution (1K), this approach does not entail a significant time cost.
>
> ### **Q3: The method requires high memory and computational cost. There are some acceleration sampling methods for diffusion or flow methods. Can they be used in the proposed method to accelerate the generation?**
> **A3:** Thanks for your insightful advice. As a training-free and model-agnostic framework, HiFlow can be seamlessly integrated with various acceleration methods for diffusion models. Here we take TeaCache [4] (an advanced caching-based acceleration technique) as an example. Equipped with TeaCache, HiFlow achieves a notable speedup in generating high-resolution images without significant quality degradation. Quantitative results are listed below:
> | Metric-4K            | FID   | FID_patch | IS    | IS_patch | CLIP Score | Latency (sec) | Speedup |
> |----------------------|:-------:|:-----------:|:-------:|:----------:|:-------------:|:----------------:|:---------:|
> | HiFlow               | 52.55 | 45.01     | 24.62 | 9.73     | 35.40       | 379            | 1.00x   |
> | HiFlow + TeaCache    | 55.48 | 52.73     | 22.94 | 8.04     | 35.26       | 212            | 1.79×   |
>
> ### **Q4: Is there any detection method to prevent errors in the alignments?**
> **A4:** Thanks for your constructive comment. **MLLM or reward models can be employed as detectors for errors in low-resolution results**. For instance, inference time scaling techniques [2] typically include an auxiliary evaluation model (such as a MLLM or a specific verifier model like ImageReward [3]) to assess the quality of the diffusion model's outputs and iteratively search for the next noise in the neighborhood of already searched high-quality noises. Therefore, these auxiliary evaluation models can serve as effective detectors for structural anomalies in the low-resolution results, thus preventing errors in the following alignments for high-resolution generation.
>
> ### **Reference**
>
> [1] Zhang L, Rao A, Agrawala M. Adding conditional control to text-to-image diffusion models[C]//Proceedings of the IEEE/CVF international conference on computer vision. 2023: 3836-3847.
>
> [2] Ma N, Tong S, Jia H, et al. Inference-time scaling for diffusion models beyond scaling denoising steps[J]. arXiv preprint arXiv:2501.09732, 2025.
>
> [3] Xu J, Liu X, Wu Y, et al. Imagereward: Learning and evaluating human preferences for text-to-image generation[J]. Advances in Neural Information Processing Systems, 2023, 36: 15903-15935.
>
> [4] Liu F, Zhang S, Wang X, et al. Timestep Embedding Tells: It's Time to Cache for Video Diffusion Model[C]//Proceedings of the Computer Vision and Pattern Recognition Conference. 2025: 7353-7363.

---

> > ### Comment · Reviewer_b5PA · 2025-08-05
> >
> > Thanks for the detailed response. The authors have adequately addressed my concerns regarding artifacts in low-resolution generation quality, computational cost, as well as error propagation and detection. Therefore, I have decided to raise my score.

---

> > > ### Author Response · Authors · 2025-08-05
> > >
> > > Dear Reviewer b5PA,
> > >
> > > Thanks for your positive feedback on our work!
> > >
> > > I'm glad to hear that our rebuttal has addressed your concerns. We truly appreciate the time and effort you've dedicated to the review process.
> > >
> > > Best regards,
> > >
> > > Authors

---

### Official Review · Reviewer_mZxP · 2025-07-06

**Clarity:** 2
**Significance:** 4
**Originality:** 4
**Rating:** 5
**Confidence:** 4

**Summary:**

- HiFlow is a training free approach for high resolution image generation using low resolution pre-trained models.
- Existing methods neglect the intermediate states of the low resolution sampling trajectory and rely only on the end points. This results in artifacts or low-fidelity details in the generated output.
- HiFlow relies on a virtual reference flow within the high-resolution space that effectively captures the characteristics of low-resolution flow information. This trajectory offers guidance for high resolution generation through 1) initialization alignment for low frequency consistency, 2) direction alignment for structure preservation and 3) acceleration alignment for detail fidelity.
- Extensive experiments show that HiFlow elevates the quality of high resolution image synthesis with the help of a well designed flow-aligned guidance.

**Questions:**

- How did the authors end up looking at the first order derivative of vector $v_t$ (i.e. acceleration alignment). The initialization and direction alignment are very intuitive but the acceleration alignment isn't. Please explain how the authors ended up investigating the acceleration. This can help understand the method even better.

**Ethical Concerns:**

["NO or VERY MINOR ethics concerns only"]

**Final Justification:**

I thank the authors for addressing my concerns thoroughly. After reading through the rebuttals and other reviews, I would like to retain my score. Authors have clearly explained the benefits of the cascaded approach compared to a direct one. They have also addressed my concerns about notation. I strongly recommend authors to add both these responses to the main paper or in the supplementary to understand the paper better.

**Limitations:**

yes.

**Paper Formatting Concerns:**

No.

**Quality:**

4

**Strengths And Weaknesses:**

**Strengths**
- HiFlow is training free and is architecture agnostic (works well with UNet or DiT family of architectures).
- Qualitative and quantitative results show clear improvements over baselines.
- All the experiments support the claims made in the paper.

**Weaknesses**
- HiFlow is a cascaded approach, so all intermediate resolutions have to be generated to reach high resolution (like 2k and 4k have to be generated to sample a 8k image).
- The notations in the paper are very hard to understand. Authors use $X_{0\leftarrow t}$ and $X_{1\leftarrow t}$ extensively in all equations throughout the paper without spending some time discussing what that quantity is. This makes it very hard to follow the rest of the paper. How can we extract $X_{0\leftarrow t}$ given the prediction of the model and how does it look for a sample generation? This is very important to understand the notation and the equations in the paper.

---

> ### Author Rebuttal · Authors · 2025-07-30
>
> We are highly encouraged that you have found our work to have significant improvements over various strong baselines. We sincerely thank you for your valuable suggestions, which certainly help improve our work. We have accordingly refined our paper as follows:
>
> ---
>
> ### **Q1: HiFlow is a cascaded approach, thus all intermediate states have to be synthesized for generating a high-resolution image.**
> **A1:** Thanks for your valuable comment. Admittedly, given the cascaded nature of HiFlow, we have to synthesize all intermediate states in the process of generating high-resolution images. However, **it can be theoretically and experimentally proven that the cascaded synthesis of high-resolution images is more efficient than direct synthesis in both time and memory**.
>
> Assuming the single-step inference time cost for generating a 1K image is $m$, due to the quadratic increase in time cost with the number of image tokens in self-attention, it can be approximated that the single-step inference time cost for generating an $n$K image is $n^2m$. Therefore, the time cost for directly sampling 30 steps to generate a 4K image can be expressed as:
>
> $$t_\text{Direct} = 30 \times 4^2\times m = 480m.$$
>
> In HiFlow, we perform the full 30-step generation at the 1K resolution, but only a portion of the timesteps at 2K and higher resolutions. According to our parameter settings, the time cost for HiFlow to synthesize a 4K image can be estimated as:
>
> $$t_\text{HiFlow} = 30 \times m + 18 \times 2^2 \times m + 9 \times 3^2 \times m + 9 \times 4^2 \times m = 327m,$$
>
> which indicates $t_\text{HiFlow} < t_\text{Direct}$, demonstrating that HiFlow is more efficient than directly inferring a 4K image. The latency comparison is provided in the following table:
>
> | Method                | Latency (sec) |
> |----------------------|:---------------:|
> | Direct Inference 4K  | 572           |
> | **HiFlow**           | **379**       |
>
> Actually, HiFlow has achieved state-of-the-art efficiency among training-free methods, as shown in Tab.2 in the main paper.
>
> Moreover, since the intermediate states are stored in the form of latent representations rather than internal model features (such as hidden states), the additional memory overhead is not significant, as shown in the following table:
>
> | Method              | Peak GPU Memory Usage (GB) |
> |---------------------|:----------------------------:|
> | Direct Inference 4K | 49.80                      |
> | HiFlow              | 50.26                      |
>
>
>
> ### **Q2: What do $X_{0\leftarrow t}$ and $X_{1\leftarrow t}$ represent, respectively? How can we extract $X_{0\leftarrow t}$ given the prediction of the model and how does it look for a sample generation?**
> **A2:** Thanks for your constructive suggestion, and we sincerely apologize for any misunderstandings that may have been caused. $X_{0\leftarrow t}$ denotes the predicted/estimated clean component of the current noisy latent $X_t$, while $X_{1\leftarrow t}$ represents its corresponding noise component at timestep $t$.
>
> For rectified flow models, the model output $v_\theta(X_t, t, c)$ stands for the flow vector/velocity (we use $v_t$ to denote $v_\theta(X_t, t, c)$ for simplicity). Given the model output $v_t$ and current noisy latent $X_t$ at timestep $t$, the estimated clean sample $X_{0\leftarrow t}$ and its corresponding estimated noise $X_{1\leftarrow t}$ can be obtained by:
>
> $$X_{0\leftarrow t} = X_t - v_tt,$$
> $$X_{1\leftarrow t} = X_t + v_t(1-t).$$
> We will include visualizations of $X_{0\leftarrow t}$ and $X_{1\leftarrow t}$ in future versions to provide further clarity. At the current stage, a visualization example of $X_{0\leftarrow t}$ can be found in our pipeline figure (e.g., the visualization of $X^\text{low}_{0\leftarrow \tau}$).
>
> ### **Q3: Motivation of acceleration alignment.**
> **A3:** Thank you for your constructive comment. Equipped with initialization alignment and direction alignment, we are able to generate high-resolution images with rich details and reasonable structures. However, it is observed that the synthesized details are often low-fidelity and unrealistic (see Fig.3 (d) in the main paper). To address these issues, we analyze the flow velocity $v_t$ at each timestep. It indicates the denoising direction of each step in the sampling trajectory, thereby determining the position of the next state. The emergence of low-fidelity details suggests that the content synthesized by the high-resolution flow deviates from the normal content synthesis distribution. This stems from the flow model's unstable performance in high-resolution generation.
>
> To this end, **we consider utilizing the rate of change of velocity (which represents acceleration in physics) to control the amount of change in velocity at each timestep within a reasonable range, thereby avoiding suboptimal denoising directions caused by unreasonable velocity predictions during high-resolution generation**. Such a guidance signal is provided by the acceleration of the reference flow at each timestep, as it reflects the detail synthesis distribution learned by the model at its training resolution.
>
> Through further derivation of the acceleration (detailed in the main paper), we uncover that the acceleration term $a_t$ can be expressed as:
> $$a_t = -\frac{1}{t_i}\frac{X_{0\leftarrow t_{i-1}}-X_{0\leftarrow t_i}}{t_{i-1}-t_i}=-\frac{1}{t_i}\frac{dX_{0\leftarrow t}}{dt},$$
> which indicates that the acceleration term $a_t$ essentially depicts the first order derivative of the estimated clean sample $X_{0\leftarrow t}$ with respect to timestep $t$, multiplied by a time-dependent factor $-1/t$. Therefore, $a_t$ can serve as an indicator of the sequence of content synthesized at each timestep, capturing the difference between estimated clean samples. In other words, $a_t$ reflects what content the model is responsible for adding at different timesteps. **Motivated by the above analysis and derivation, we adopt acceleration alignment in high-resolution generation to facilitate higher detail fidelity**.

---

> > ### Author Response · Authors · 2025-08-07
> >
> > Dear Reviewer mZxP,
> >
> > We sincerely appreciate your time and effort in providing such meticulous reviews and insightful comments.
> >
> > Could you please kindly let us know if our rebuttal has addressed your concerns?
> >
> > If you have any further questions regarding our work, we would be delighted to address them.
> >
> > Best regards,
> >
> > Authors

---

### Note · Authors · 2025-08-11

We sincerely appreciate ACs, SACs, PCs and all reviewers for their time and efforts in the review. We received positive ratings from all five reviewers in the preliminary review, which have been very motivating for us. We are highly encouraged that: Reviewer mZxP found our work **to have significant improvements over various strong baselines**; Reviewer b5PA described our work as **reasonable, interpretable and effective**; Reviewer 7Jea found our work **innovative, sound and addressing a significant challenge in text-to-image generation**; Reviewer htDR described our work **effective, reasonable, and easy to understand**; Reviewer iWGG considered our work **effective and conducted comprehensive experiments**. Based on their valuable and insightful comments, we have carefully revised our manuscript, as detailed below:

1. We have added **a more detailed analysis and discussion on the flow acceleration**. Specifically, we have elaborated on the motivation of studying acceleration, the effects of acceleration alignment and the physical significance of acceleration visualization.

2. We have supplemented **comprehensive sensitivity analysis experiments** regarding guidance weight strategies and hyperparameters across various resolutions.

3. We have discussed **the robustness of our method** and provided **mitigation strategies for potential structural flaws**.

4. We have **integrated our method with the acceleration technique TeaCache** to offer improved efficiency.

5. We have validated the potential of the proposed **HiFlow in the leading video generation model WAN2.1**, in which HiFlow demonstrates effectiveness in training-free high-resolution video generation with flow-aligned guidance.

6. We have conducted **quantitative experiments on a new benchmark** (LAION-5B dataset) to more comprehensively demonstrate the effectiveness of HiFlow.

7. We have **clarified ambiguous expressions in the paper**, including the meanings of notations, our evaluation dataset, and the components of the framework figure.

8. We have proven that **the cascaded architecture of HiFlow does not entail additional time cost** and **HiFlow barely increase GPU memory usage** compared to directly synthesizing high-resolution images.

During the rebuttal period, we once again received positive feedback and recognition from the reviewers. We will incorporate the above revisions into the final version. Thanks again to ACs, SACs, PCs and all the reviewers for their contributions.

---

### Decision · Program_Chairs · 2025-09-17

**Decision:**

Accept (poster)

**Comment:**

The paper introduces HiFlow, a training-free model-agnostic framework for generating high-resolution images using a pre-trained text-to-image flow model trained on lower-resolution images. This is achieved by constructing a virtual reference flow in the high-resolution space that effectively captures low-resolution flow information to guide the high-resolution sampling process. This reference flow incorporates three sources of guidance: initialization alignment, direction alignment, and acceleration alignment. This three-pronged approach is improvement over an existing method that incorporates the low-resolution information only for the endpoint of the high-resolution flow trajectory, which can result in image artifacts.

The empirical evaluation shows that the method outperforms the existing training-free methods and is also more efficient. The authors' claim based on a handful of samples that their approach also outperforms training-based methods is however not convincing and should be removed.

Among the strength of the paper are:
- A novel and effective algorithm for generating high-resolution images based on a lower-resolution flow model.
- The algorithm is both training-free and model-agnostic, which means it's widely applicable and will benefit from improvements to the base models.
- The empirical results are impressive.
- The ablation study is helpful in understanding the effect of each of the sources of guidance

The paper does have a handful of weaknesses:
- The empirical evaluation used just a single dataset, though the authors provided results on another dataset in response to the reviewers.
- It is not easy to understand how the three sources of guidance fed into the sampler, and the figure describing the method is rather complicated, though perhaps not unnecessarily. The authors are encouraged to include the pseudo-code for the method in the supplementary material, it would be easier to follow than the included source code.
- Reviewers found acceleration guidance less clearly motivated than the other two guidance sources.

The reviewers generally liked the paper and the vast majority of their concerns were addressed by the authors. This paper makes a solid contribution to the field and should be accepted.